# Distinct molecular and immune hallmarks of inflammatory arthritis induced by immune checkpoint inhibitors for cancer therapy

Sang T. Kim [1,21], Yanshuo Chu[2,21], Mercy Misoi[3], Maria E. Suarez-Almazor[1], Jean H. Tayar[1], Huifang Lu[1], Maryam Buni[1], Jordan Kramer[4,5], Emma Rodriguez[6], Zulekha Hussain[1], Sattva S. Neelapu [7], Jennifer Wang[8], Amishi Y. Shah [8], Nizar M. Tannir[8], Matthew T. Campbell [8], Don L. Gibbons [9], Tina Cascone [9], Charles Lu[9], George R. Blumenschein[9], Mehmet Altan[9], Bora Lim[10], Vincente Valero[10], Monica E. Loghin[11], Janet Tu [12], Shannon N. Westin [13], Aung Naing [14], Guillermo Garcia-Manero [15], Noha Abdel-Wahab[1,16,17], Hussein A. Tawbi [16], Patrick Hwu[16,20], Isabella C. Glitza Oliva[16], Michael A. Davies [16], Sapna P. Patel [16], Jun Zou [18], Andrew Futreal [2], Adi Diab[16], Linghua Wang [2,19✉] & Roza Nurieva [4,19✉]

Immune checkpoint inhibitors are associated with immune-related adverse events (irAEs), including arthritis (arthritis-irAE). Management of arthritis-irAE is challenging because immunomodulatory therapy for arthritis should not impede antitumor immunity. Understanding of the mechanisms of arthritis-irAE is critical to overcome this challenge, but the pathophysiology remains unknown. Here, we comprehensively analyze peripheral blood and/or synovial fluid samples from 20 patients with arthritis-irAE, and unmask a prominent Th1-CD8$^+$ T cell axis in both blood and inflamed joints. CX3CR1$^{hi}$ CD8$^+$ T cells in blood and CXCR3$^{hi}$ CD8$^+$ T cells in synovial fluid, the most clonally expanded T cells, significantly share TCR repertoires. The migration of blood CX3CR1$^{hi}$ CD8$^+$ T cells into joints is possibly mediated by CXCL9/10/11/16 expressed by myeloid cells. Furthermore, arthritis after combined CTLA-4 and PD-1 inhibitor therapy preferentially has enhanced Th17 and transient Th1/Th17 cell signatures. Our data provide insights into the mechanisms, predictive biomarkers, and therapeutic targets for arthritis-irAE.

---

*A list of author affiliations appears at the end of the paper.

Understanding the cellular and molecular basis of T cell activation paved the way to the development of immune checkpoint inhibitors (ICIs), one of the greatest achievements in the life sciences[1]. Full activation of T cells requires two signals: (1) binding of the T cell receptor (TCR) to the antigen presented by the major histocompatibility complex on antigen-presenting cells; and (2) co-stimulation by engagement of CD80/86 on antigen-presenting cells to CD28 on T cells[1]. Once fully activated, the T cells undergo robust proliferation and clonal expansion. At some point after activation, to control hyper-activation and prevent unnecessary tissue injury, activated T cells express negative immune regulators on their surface[1]. Cytotoxic T lymphocyte-associated protein 4 (CTLA-4) is a receptor that blocks engagement of CD28-CD80/86 via its high affinity to CD80/86. Programmed cell death protein 1 (PD-1) is another negative immune regulator that down-modulates TCR signaling. By inhibiting CTLA-4 or PD-1, and rejuvenating antitumor immunity, ICIs have shown groundbreaking success in cancer treatment[1].

Despite unprecedented clinical success, ICIs can lead to life and/or organ-threatening immune-related adverse events (irAEs)[1]. A systematic review revealed that irAE profiles differ by ICI regimen[2]. CTLA-4 inhibitor therapy is more frequently associated with colitis, hypophysitis, and skin toxicity. In contrast, pneumonitis, hypothyroidism, and vitiligo are more common with PD-1 inhibitor therapy.

Arthritis is the most common rheumatic irAE, occurring in about 5% of patients receiving an ICI[3]. Arthritis-irAE is associated with either PD-1 inhibitor monotherapy or combination of CTLA-4 and PD-1 inhibitor therapy[3]. Although most irAEs develop within 12 weeks after the first infusion of an ICI(s)[4], arthritis-irAE can occur later[3]. Arthritis-irAE can greatly impair quality of life, lead to discontinuation of ICI(s) therapy, and/or cause bony destruction due to its inflammatory nature[5]. Mild arthritis-irAE is initially treated with nonsteroidal anti-inflammatory drugs. If arthritis-irAE becomes moderate or severe, corticosteroids are initiated and ICI(s) may need to be withheld. If the arthritis-irAE does not respond to corticosteroids or the steroids cannot be tapered off (termed steroid resistance), steroid-sparing disease-modifying anti-rheumatic drugs (DMARDs) are administered[6]. Of note, recent studies revealed that steroids significantly abrogate the antitumor efficacy of ICIs at a dose of ≥10 mg daily prednisone or equivalent[7,8]. This is particularly important in managing arthritis-irAE because arthritis-irAE may have a chronic course, requiring long-term immunosuppression for months or even years[3].

To formulate an optimal therapeutic strategy without impeding antitumor immunity, it is critical to elucidate the underlying mechanisms of arthritis-irAE. Elucidating the mechanisms of arthritis-irAE has been challenging mainly because, as with other irAEs, there are no preclinical mouse models recapitulating arthritis-irAE[9]. Therefore, comprehensive analysis of patients' biospecimens and clinical data is critical to get mechanistic insights of arthritis-irAE.

In this work, we perform integrative immunophenotyping of peripheral blood (PB) and synovial fluid (SF) from patients with arthritis-irAE utilizing single-cell RNA sequencing (scRNAseq), single-cell TCR sequencing (scTCRseq), flow cytometry, in vitro functional assays, and cytokine multiplex assay. We also prospectively follow the patients for 12 months to determine arthritis-irAE clinical outcomes and response to irAE therapy. We identify molecular and cellular hallmarks in arthritis-irAE, which are distinct from those in classical autoimmune arthritis or non-arthritic irAEs. Our analyses reveal that Th1-CD8$^+$ T cells play a pivotal role in arthritis-irAE disease pathogenesis, and

Th17 cells and transient Th17 cells, enriched in arthritis after combined CTLA-4 and PD-1 inhibitor therapy, are involved in steroid resistance. Our discovery will enable us to identify predictive biomarkers and therapeutic targets for arthritis-irAE, an emerging but clinically unmet disease entity.

## Results

**Cohort of arthritis-irAE to perform clinical, molecular, and immunologic analyses.** From July 2017 to June 2019, we recruited 20 patients who newly developed arthritis after ICI therapy (Fig. 1a). The diagnosis of inflammatory arthritis was determined by a history and physical examination performed by a treating rheumatologist (S.T.K., M.S.-A., J.H.T, and H.L.) at The University of Texas MD Anderson Cancer Center (UTMDACC). Arthritis-irAE was treated following the guidelines from The Society for Immunotherapy of Cancer[6]. In addition, from the study enrollment, we followed the patients for 12 months prospectively to monitor whether steroid monotherapy failed and if the patient needed DMARDs as a steroid-sparing agent.

Along with standard-of-care laboratory analysis, residual PB and/or SF samples were collected at the time of the active arthritis (Supplementary Fig. 1a, b). We also collected longitudinal PB samples after treatment for arthritis-irAE from seven patients who were not receiving steroids during the initial sample collection and agreed to donate longitudinal PB samples (Supplementary Fig. 1a, b). We characterized altered immunity of arthritis-irAE by analyzing PB and SF samples with scRNAseq, scTCRseq, flow cytometry, cytokine multiplex assay on PB and SF samples, and in vitro functional assays to interrogate the immunobiology of arthritis-irAE (Supplementary Fig. 1b). As a negative control in serum analyses, we collected PB samples from patients who had not developed irAEs at least 12 weeks after initiating ICI therapy (no-irAE group; n = 9). Because the onset of the arthritis-irAE varied in our cohort (range: 0.5–365 weeks), we choose 12 weeks as a cutoff because >90% of patients develop irAEs within 12 weeks of initiating ICI therapy[6]. As a negative control in SF supernatant analyses, we collected SF samples from ICI-naïve patients with osteoarthritis (osteoarthritis group; n = 4). The patients met the American College of Rheumatology diagnostic criteria for osteoarthritis[10]. The study was approved by the institutional review board at UTMDACC. Patients provided written informed consent allowing the use of their biospecimens (IRB No: PA16-0935).

**Clinical and sample characteristics.** Clinical characteristics of the patients are summarized in Table 1 and Supplementary Data 1. Melanoma was the most common malignancy (11/20) in our cohort. Eleven patients received PD-1 inhibitor monotherapy while 9 patients received both CTLA-4 and PD-1 inhibitors (sequential ICI therapy [n = 5]; combination ICI therapy [n = 4]). We previously described several patterns of arthritis in arthritis-irAE[11]. All of our patients had undifferentiated inflammatory arthritis with oligoarthritis (n = 13; mainly involving knees and/or ankles), polyarthritis (n = 4; mainly involving hands and/or wrists), or monoarthritis (n = 3; all unilateral knee). No patients had uveitis or enthesitis associated with arthritis. Only one patient had HLA-B27. Sixty percent of the patients had other irAEs with CTCAE grade ≥2 prior to the arthritis, and colitis-irAE (35%) was the most common of these followed by dermatitis (20%). Consistent with other studies[12–14], arthritis in our cohort generally developed more than 12 weeks after the first ICI infusion; however, the onset of arthritis-irAE varies with range from 0.5 to 365 weeks from the first ICI infusion. Seven patients developed arthritis after stopping ICI therapy. The clinical disease activity index (CDAI) is calculated as the summation of patient

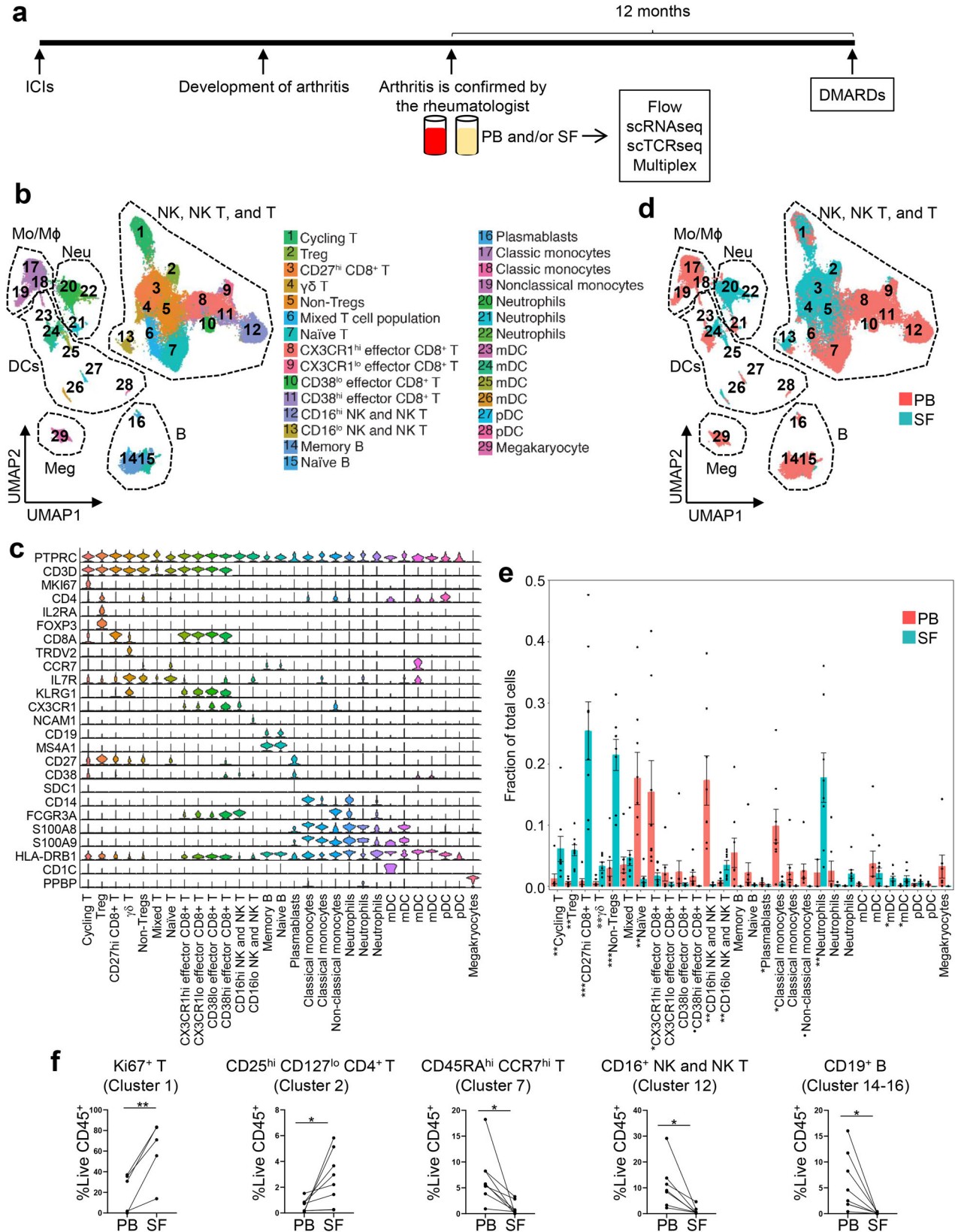

global assessment (0–10), physician global assessment (0–10), and swollen/tender joint counts of 28 joints[15]. CDAI ranges from 0 to 72, with scores representing remission (0–2.8), low (2.9–10.0), moderate (10.1–22.0), and high (>22) arthritis activity. The median CDAI of our patients was 21.5, suggesting high arthritis disease activity. In addition, 15% of patients with arthritis-irAE

tested positive for anti-nuclear antibody and 10% tested positive for rheumatoid factor. No patients tested positive for anti–cyclic citrullinated peptide antibody.

At the time of active arthritis-irAE, we collected paired PB and SF from 8 patients, PB only from 9 patients, and SF only from 3 patients (Supplementary Data 1; Supplementary Table 1;

**Fig. 1 Peripheral blood (PB) and synovial fluid (SF) exhibit distinctive immune cell landscapes in arthritis as an immune-related adverse event.**
**a** Project outline. Created by authors using PowerPoint. ICI, immune checkpoint inhibitor; scRNAseq, single cell RNA sequencing; scTCRseq, single cell T cell receptor sequencing; DMARD, disease-modifying anti-rheumatic drug. **b** UMAP view of 6 major cell lineages and 29 cell subsets. Each dot represents a single cell that is color-coded by cluster ID. Mo, Monocytes; Mϕ, macrophages; NK, natural killer cells; Neu, neutrophils; DCs, dendritic cells; Meg, megakaryocytes; B, B cells; T, T cells; Treg, regulatory T cells; mDC, myeloid dendritic cells; pDC, plasmacytoid dendritic cells. **c** Gene expression of canonical markers across clusters. See Supplementary Data 2 for all differentially expressed genes. **d** Identification of cell clusters distinctly present in PB or SF. The cells are color-coded by their sample origins. **e** Quantification of cell clusters and comparison of the cellular fraction of each cluster between PB ($n = 8$) and matching SF samples. Two-sided unpaired $t$ test. Cycling T cells, **$P = 0.009$; Tregs, **$P = 0.0014$; CD27$^{hi}$ CD8$^+$ T cells, ***$P = 0.0008$; γδ T cells, **$P = 0.004$; non-Tregs, ***$P = 0.0007$; naïve T cells, **$P = 0.007$; CX3CR1$^{hi}$ effector CD8$^+$ T cells, *$P = 0.038$; CD38$^{hi}$ effector CD8$^+$ T cells, •$P = 0.059$; CD16$^{hi}$ natural killer (NK)/NK T cells, **$P = 0.003$; CD16$^{lo}$ NK/NK T cells, **$P = 0.003$; Plasmablasts, *$P = 0.032$; Classical monocytes (left), *$P = 0.011$; Non-classical monocytes, •$P = 0.064$; neutrophils (left), **$P = 0.0055$; myeloid dendritic cells (mDCs; third from the left), *$P = 0.011$; mDC (fourth from the left), *$P = 0.043$. Bars indicate the mean and SEM. **f** Flow cytometry analysis to quantify proportions of Ki67$^+$ T cells ($n = 5$), CD25$^{hi}$ CD127$^{lo}$ CD4$^+$ cells ($n = 8$), CD45RA$^{hi}$ CCR7$^{hi}$ T cells ($n = 8$), CD16$^{hi}$ NK and NK T cells ($n = 7$), and CD19$^+$ B cells ($n = 8$) from PB and matching SF samples. Two-sided paired $t$ test. (left to right) **$P = 0.0068$, *$P = 0.0280$, *$P = 0.0164$, *$P = 0.0237$, *$P = 0.0214$. Source data are provided as a Source Data file.

Supplementary Fig. 1b). Samples were collected at a median of 45.8 weeks after the first ICI infusion and 8.5 weeks after the development of the arthritis. Eight of 20 patients (40%) with arthritis-irAE had received systemic steroids within 4 weeks prior to the sample donation, and two patients were receiving systemic steroids at the time of the sample donation. Regarding SF, median white blood cell counts were 4800 cells/μL (range 774–46,360 cells/μL). Neutrophils were the predominant cell population (median 75.0%), followed by lymphocytes (median 13.5%) and histiocytes (10.5%).

**PB and SF contain distinctive immune cell subsets.** As the first step to elucidate the mechanisms of arthritis-irAE, we aimed to capture the global immune cell landscape and the functional states of immune cells in arthritis-irAE. To achieve these aims, we performed 5' barcoded scRNAseq to analyze paired PB and SF samples from eight patients with arthritis-irAE (Fig. 1; Supplementary Fig. 1). scRNAseq data was processed and we retained 89,785 cells (59,323 from PB and 30,462 from SF samples) following rigorous quality control. Unsupervised clustering on the basis of transcriptome similarity identified 29 cell clusters (Fig. 1b–e; Supplementary Data 2). Cells from each PB or SF samples were distributed across the clusters, indicating minimal batch effects (Supplementary Fig. 1c). Compared to PB, clusters expanded in SF included: cycling T cells (cluster 1), Tregs (cluster 2), CD27$^{hi}$ CD8$^+$ T cells, (cluster 3), γδ T cells (cluster 4), non-Tregs (clusters 5), CD16$^{lo}$ natural killer (NK)/NK T cells (cluster 13), CD16$^{hi}$ neutrophils (cluster 20), and myeloid dendritic cells (clusters 25–26) (Fig. 1d, e). In contrast, naïve T cells (cluster 7), CX3CR1$^{hi}$ effector CD8$^+$ T cells (cluster 8), CD38$^{hi}$ effector CD8$^+$ T cells (cluster 11), plasmablasts (cluster 16), and nonclassical monocytes (cluster 19) were more abundant in PB (Fig. 1d, e).

To validate scRNAseq findings, we enumerated major immune cell subsets of the SF and PB samples with flow cytometry using the gating strategy as we previously described (Supplementary Fig. 1d)[16]. Consistently, flow cytometry analyses revealed that the frequencies of Ki67$^+$ T cells (analogue to cluster 1) and CD25$^{hi}$ CD127$^{lo}$ Tregs (cluster 2) were significantly higher in SF while the frequencies of CD45RA$^{hi}$ CCR7$^{hi}$ CD3$^+$ naïve T cells (cluster 7), CD16$^{hi}$ NK and NK T cells (cluster 12), and CD19$^+$ B cells (clusters 14–16) were higher in PB (Fig. 1f; Supplementary Fig. 1e). Likewise, flow cytometry analyses suggested a trend towards increased proportions of CD27$^{hi}$ CD8$^+$ T cells (cluster 3), γδ T cells (cluster 4), and CD16$^{hi}$ neutrophils (cluster 20) in SF and increased proportion of CD27$^{lo}$ CD8$^+$ T cells (clusters 8–11) in PB samples.; however, the differences in immune cell subsets between PB and SF T cells did not reach statistical significance (Supplementary Fig. 1e, f).

Collectively, our scRNAseq analyses revealed major immune cell clusters, including those distinctly present in SF and PB. Furthermore, we validated the scRNAseq findings in protein levels utilizing flow cytometry.

**Interferon gamma (IFNγ)-producing Th1/Tc1 cells might play a role in arthritis-irAE according to subcluster analysis of NK, NK T, and T cell clusters.** Immune checkpoint inhibitors mainly target T cells[17], and our scRNAseq data revealed the presence of distinct clusters of NK, NK T, and T cells (Fig. 1). Therefore, we focused more on NK, NK T, and T cell clusters and performed subclustering analysis (Fig. 2). We identified 17 subclusters, including naïve T cells, activated and memory T cells, Tregs, cycling T cells, effector CD8$^+$ T cells, mucosal-associated invariant T cells (MAIT), γδ T cells, NK cells, and NK T cells (Fig. 2a, b; Supplementary Data 3). Naïve T cells (subclusters 1–2 and 6), CX3CR1$^{hi/lo}$ effector CD8$^+$ T cells (subclusters 10–11), CX3CR1$^{hi}$ γδ T cells (subcluster 15), and CD16$^{hi}$ NK/NK T cells (subcluster 16) were enriched in PB (Fig. 2c, d). In contrast, CXCR3$^{hi}$ Th1-like and *KLRB1* (encoding CD161)$^{hi}$ Th17-like cells (subcluster 3), Tregs (subcluster 4), PD-1$^{hi}$ CXCL13$^+$ T cells (subcluster 5), CXCR3$^{hi}$ effector CD8$^+$ T cells (subclusters 8–9), cycling T cells (subcluster 12), mucosal-associated invariant T cells (subcluster 13), CX3CR1$^{lo}$ γδ T cells (subcluster 14), and CD16$^{lo}$ NK/NK T cells (subcluster 17) were enriched in SF (Fig. 2c, d).

In each cluster, we compared gene expression of key effector T cell cytokines, including IFNγ (Th1/Tc1 cells), interleukin (IL)-4 (Th2/Tc2 cells), IL-17 (Th17/Tc17 cells), IL-10 (Tregs), and IL-21 (follicular helper T cells)[18]. Among genes encoding these T cell effector cytokines, the *IFNG* gene (encoding IFNγ) was most abundantly expressed by NK, NK T, and T cell subclusters (Fig. 2e). Intracellular staining of PB and SF samples, performed after stimulation of T cells with phorbol 12-myristate 13-acetate (PMA) and ionomycin for 4 h, also demonstrated an abundance of IFNγ-producing T cells (Fig. 2f), validating scRNAseq findings. Genes encoding inflammatory molecules such as tumor necrosis factor alpha (*TNF*), granzyme B (*GZMB*), granzyme K (*GZMK*), granzyme H (*GZMH*), granulysin (*GNLY*), and GM-CSF (*CSF2*) were highly expressed in all 17 subclusters (Fig. 2e).

Together, our NK, NK T, and T cell subcluster analyses revealed that IFNγ-producing Th1/Tc1 cells might play a critical role in the pathogenesis of arthritis-irAE.

**Tregs are enriched in SF and have enhanced suppressive functions.** Given the important role of Tregs in autoimmune diseases[19], we further explored the transcriptomic landscape of Tregs (Fig. 3). We identified two subclusters of Tregs; one was

**Table 1 Demographic and clinical characteristics of the patients in our study[a].**

| Characteristic | No. (%) | | |
|---|---|---|---|
| | Arthritis-irAE[a], *n* = 20 | Osteoarthritis, *n* = 4 | No-irAEs, *n* = 9 |
| Median age (range) | 56.5 years (34-77) | 60.5 years (57-64) | 62 years (47-77) |
| Male | 14 (70) | 2 (50) | 5 (56) |
| Median BMI (range) | 28.7 (21.7-37.9) | 31.2 (29.6-31.6) | 30.8 (22.4-33.7) |
| Tumor type | | | |
| Melanoma | 11 (55) | 1 (25) | 6 (67) |
| RCC | 4 (20) | 0 (0) | 0 (0) |
| NSCLC | 4 (20) | 1 (25) | 3 (33) |
| Neuroendocrine | 1 (5) | 2 (50) | 0 (0) |
| ICI(s) | | | |
| Nivolumab | 6 (30) | N/A | 6 (67) |
| Pembrolizumab | 4 (20) | N/A | 1 (11) |
| Durvalumab | 1 (5) | N/A | 1 (11) |
| Ipilimumab + nivolumab | 6 (30) | N/A | 1 (11) |
| Ipilimumab + pembrolizumab | 3 (15) | N/A | 0 (0) |
| CTCAE[c] grade II–IV irAEs prior to the arthritis | | | |
| Colitis | 7 (35) | N/A | N/A |
| Dermatitis | 4 (20) | N/A | N/A |
| Myocarditis | 1 (5) | N/A | N/A |
| Type I diabetes mellitus | 1 (5) | N/A | N/A |
| Hypothyroidism | 1 (5) | N/A | N/A |
| Neuritis | 1 (5) | N/A | N/A |
| Pattern of arthritis | | | |
| Undifferentiated monoarthritis | 3 (15) | N/A | N/A |
| Undifferentiated oligoarthritis | 13 (65) | N/A | N/A |
| Undifferentiated polyarthritis | 4 (20) | N/A | N/A |
| Median time from first ICI infusion to development of arthritis (range) | 35.5 weeks (0.5–365) | N/A | N/A |
| Stopped ICI therapy before development of arthritis | 6 (30) | N/A | N/A |
| Colitis-irAE | 1 (5) | N/A | N/A |
| Myocarditis-irAE | 1 (5) | N/A | N/A |
| Neuritis-irAE | 1 (5) | N/A | N/A |
| Tumor progression | 2 (10) | N/A | N/A |
| Completion of ICI therapy | 2 (10) | N/A | N/A |
| Arthritis-irAE CTCAE grade | | N/A | N/A |
| II | 16 (80) | N/A | N/A |
| III | 4 (20) | N/A | N/A |
| Median CDAI[d] (range) | 21.5 (7–60) | N/A | N/A |
| Median ESR[e] (range) | 53 mm/h (8–108) | 8 mm/h (7–9) | N/A |
| Median CRP[f] (range) | 80.0 mg/L (2.0–300) | 3.5 mg/L (1.3–5.8) | N/A |
| ANA-positive[g] | 3 (16) | N/A | N/A |
| RF-positive | 2 (10) | N/A | N/A |
| Anti-CCP antibody–positive[g] | 0 (0) | N/A | N/A |
| HLA B27-positive[g] | 1 (5) | N/A | N/A |

[a]Abbreviations: *irAEs* immune-related adverse events, *BMI* body mass index, *RCC* renal cell carcinoma, *NSCLC* non-small cell lung carcinoma, *ICI* immune checkpoint inhibitor, *N/A* not applicable, *CTCAE* common terminology criteria for adverse events, *CDAI* clinical disease activity index, *ESR* erythrocyte sedimentation rate, *CRP* C-reactive protein, *ANA* anti-nuclear antibody, *RF* rheumatoid factor, *CCP* cyclic citrullinated peptide.
[b]Other cancers included ovarian cancer (*n* = 1), breast cancer (*n* = 2), glioblastoma (*n* = 1), cervical cancer (*n* = 1), and neuroendocrine tumor (*n* = 1).
[c]CTCAE grades range from I to V, with higher values indicating more severe.
[d]Values of CDAI range from 0 to 76, with higher values indicating more active inflammatory arthritis.
[e]Normal values of ESR range from 0 to 20 mm/h.
[f]Normal values of CRP range from 0 to 10 mg/L.
[g]*n* = 19 in the arthritis-irAE group.

composed of cells mainly from PB and the other was nearly exclusively enriched in SF (Fig. 3a–c; Supplementary Data 4). We identified 52 differentially expressed genes (DEGs) between the two subclusters ($\log_2$ fold changes >0.5 or < −0.5; $-\log_{10} P$ > 15), six of which were more expressed in the PB Tregs (subcluster 1) and 46 of which were more expressed in the SF Tregs (subcluster 2) (Fig. 3d; Supplementary Data 5). Effector Tregs (eTregs), defined as CD3[+] CD4[+] CD25[hi] CD127[lo] CD45RA[−], are known to have enhanced suppressive activity in mice and humans[20], and compared with PB

Tregs, SF Tregs showed higher expression of genes related to eTreg differentiations (*TNFRSF4, TNFRSF18*), eTreg transcriptional regulations (*BATF*), Treg master regulation (*FOXP3*), and Treg suppressive functions (*CTLA4, IL2RA, TIGIT, LAG3*)[21–23]. In contrast, PB Tregs highly expressed genes relevant to early eTreg differentiations or Treg homeostasis, including *LEF1, TXNIP*, and *JUNB*[24,25]. Flow cytometry analyses validated scRNAseq findings showing that frequencies of CD25[hi] CD127[lo] CD45RA[−] eTregs were increased in SF (% of eTreg; mean ± SD; PB vs. SF;

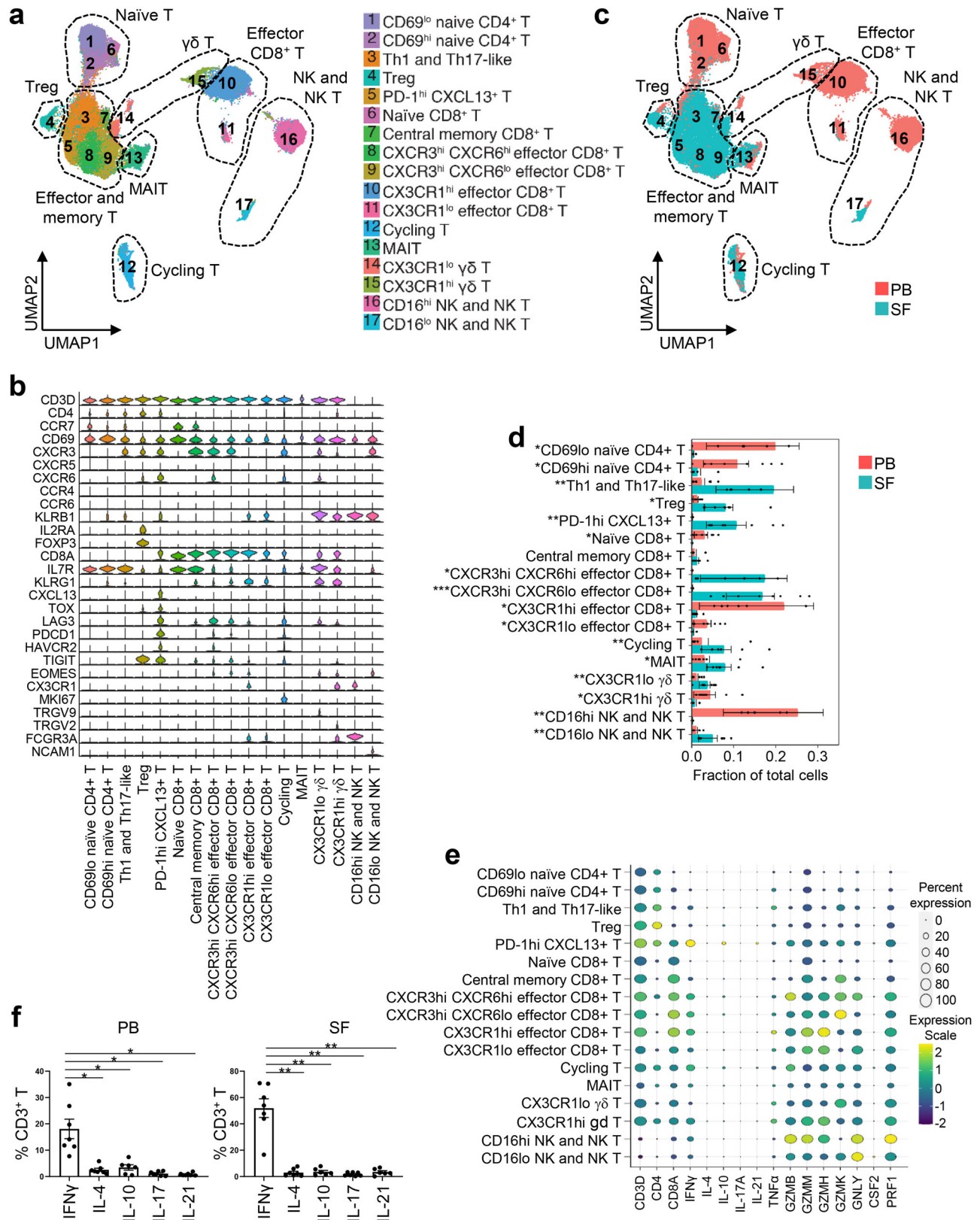

64.82 ± 17.60 vs. 97.85 ± 1.33; $P = 0.03$) (Fig. 3e). Consistent with scRNAseq findings, the mean fluorescence intensity of key Treg effector molecules, including CD25, FOXP3, and CTLA-4, was higher in SF Tregs than in PB Tregs (Fig. 3f). A recent study revealed that PD-1[hi] Tregs have enhanced suppressor functions compared with PD-1[lo] Tregs[26], and we observed that PD-1 was

more expressed on SF Tregs than on PB Tregs (Fig. 3f). scRNAseq and flow cytometry analysis of Tregs led us to hypothesize that SF Tregs have enhanced suppressive functions compared with PB Tregs. To address the hypothesis, we performed the in vitro Treg suppressor assay in paired PB and SF samples with adequate cell numbers from five patients in the arthritis-irAE group. CD45RA[hi]

**Fig. 2 Interferon gamma (IFNγ)-producing Th1/Tc1 cells might play a role in arthritis as an immune-related adverse event, according to subcluster analysis of T, natural killer (NK), and NK T cells. a** Identification of 17 subclusters of NK, NK T, and T cell populations across all samples and annotation of the subclusters. Treg, regulatory T cells; MAIT, mucosal-associated invariant T cells. **b** Gene expression of canonical markers across subclusters. See Supplementary Data 3 for all differentially expressed genes. **c** Identification of subclusters distinctly present in PB or SF. **d** Quantification of the subclusters and comparison of the cellular fraction of each cluster between PB and SF samples. $n = 8$ PB and matching SF samples. Two-sided unpaired $t$ test. CD69$^{lo}$ naïve CD4$^+$ T cells, $*P = 0.011$; CD69$^{hi}$ naïve CD4$^+$ T cells, $*P = 0.016$; Th1 and Th17-like, $**P = 0.009$; Treg, $*P = 0.011$; PD-1$^{hi}$ CXCL13$^+$ T cells, $**P = 0.003$; Naïve CD8$^+$ T cells, $*P = 0.010$; CXCR3$^{hi}$ CXCR6$^{hi}$ effector CD8$^+$ T cells, $*P = 0.014$; CXCR3$^{hi}$ CXCR6$^{lo}$ effector CD8$^+$ T cells, $***P = 0.0008$; CX3CR1$^{hi}$ effector CD8$^+$ T cells, $*P = 0.021$; CX3CR1$^{lo}$ effector CD8$^+$ T cells, $*P = 0.020$; Cycling T, $**P = 0.0012$; MAIT, $*P = 0.028$; CX3CR1$^{lo}$ γδ T cells. $**P = 0.005$; CX3CR1$^{hi}$ γδ T cells, $*P = 0.022$; CD16$^{hi}$ NK/NK T cells, $**P = 0.004$; CD16$^{lo}$ NK/NK T cells, $**P = 0.003$. Bars indicate the mean and SEM. **e** Bubble plots showing expression levels and frequencies of key effector T cell cytokines and inflammatory molecules across subclusters. $n = 8$ PB and matching SF samples. **f** Flow cytometry analysis to quantify cytokine-producing T cells after stimulation with PMA and ionomycin for 4 h. $n = 6$ (PB IL-10; SF IL-10; SF IL-21) or $n = 7$ (PB IFNγ; PB IL-4; PB IL-17; PB IL-21; SF IFNγ; SF IL-4; SF IL-17). One-way analysis of variance. $*P < 0.05$, $**P < 0.01$. PB (left to right): $*P = 0.0245$, $*P = 0.0267$, $*P = 0.0102$, $*P = 0.0125$. SF (left to right): $**P = 0.0011$, $**P = 0.0041$, $**P = 0.0013$, $**P = 0.0030$. Bars indicate the mean and SEM. Source data are provided as a Source Data file.

CCR7$^{hi}$ naïve CD4$^+$ T cells (10,000 cells/well) were co-cultured with autologous SF Tregs or PB Tregs (20,000 cells/well) in the presence of polyclonal TCR stimulations in vitro (Fig. 3g; Supplementary Fig. 2). Four days later, we measured proliferation and IFNγ and IL-2 production in CD4$^+$ T cells. As a control, we performed a parallel experiment with naïve CD4$^+$ T cells and PB Tregs from age-, sex-, tumor-, and ICI-matched patients in the no-irAE group. SF Tregs showed enhanced suppressive functions compared with PB Tregs in terms of cell proliferation as well as cytokine production (IFNγ and IL-2) (Fig. 3g; Supplementary Fig. 2).

Taken together, our data suggested that Tregs were enriched in SF of patients with arthritis-irAE, with eTreg phenotypes and enhanced suppressive functions.

**CXCR3$^{hi}$ CXCR6$^{hi/lo}$ effector CD8$^+$ T cells, recruited via CXCL9/10/11 and CXCL16 secreted by myeloid cells, play an important role in arthritis-irAE.** Next, we investigated the clonotypes and trafficking of T cells. First, we investigated the top 100 most expanded T cell clones in PB and SF (Fig. 4a). In PB, CX3CR1$^{hi}$ effector CD8$^+$ T cells (cluster 10) dominantly contributed to the top 100 T cell clones. In contrast, cells from synovial-enriched clones were mainly part of CXCR3$^{hi}$ CXCR6$^{hi/lo}$ effector CD8$^+$ T cells (clusters 8–9). Taken together, TCR analyses showed that effector CD8$^+$ T cells potentially contribute the pathogenesis of arthritis-irAE. Next, we addressed the question of whether T cell clones are shared between PB and SF by analyzing TCR clones in scTCRseq (Fig. 4b). We observed that 10,384 TCR clones were shared by at least two subclusters. Notably, 343 T cell clones were shared between CX3CR1$^{hi}$ effector CD8$^+$ T cells (cluster 10), a uniquely expanded T cell subcluster in PB, and CXCR3$^{hi}$ CXCR6$^{lo}$ effector CD8$^+$ T cells (cluster 9), one of the top two expanded cell clusters in SF, suggesting that active migration and transition occurred between the most expanded T cell clones in PB and SF. To determine whether a progeny-progenitor relationship was present between these two cell subclusters, we investigated the proportions of naïve, central memory, effector memory, and terminally differentiated effector memory cells in PB and SF (Supplementary Fig. 3a, b)[27]. Both SF CD4$^+$ T cells and CD8$^+$ T cells were mostly either effector memory or terminally differentiated effector memory cells (mean ± SD 72.96 ± 21.25 in SF CD4$^+$T cells, 76.42 ± 20.63 in SF CD8$^+$ T cells). Considering that effector memory or terminally differentiated effector memory CD8$^+$ T cells are localized to the peripheral tissues during and at the end of the inflammation[27–30], CX3CR1$^{hi}$ effector CD8$^+$ T cells in PB are thought to be the progenitor of CXCR3$^{hi}$ effector CD8$^+$ T cells in SF. We investigated the trafficking of T cells by analyzing chemokine ligands and their receptors (Fig. 4c; Supplementary Figs. 3c–4). Global analysis of all chemokine receptor-ligand pairs revealed that chemokine receptors including CCR1/7/9, CXCR3/6, SDC1, XCR1, and ACKR4 were highly expressed on T cells, and of note, SF T cells abundantly expressed CXCR3 and CXCR6 (Supplementary Figs. 3c–4). Given the important role of CXCR3 and CXCR6 in autoimmune arthritis[31] and their prominent expression on SF T cells, we focused more on CXCR3 and CXCR6, and their ligands, CXCL9/10/11 and CXCL16. We observed strong interactions between CXCR3 on T cells and CXCL9/10/11 on myeloid cells, as well as between CXCR6 on T cells and CXCL16 on myeloid cells (Fig. 4c; Supplementary Fig. 4), implying that myeloid cells might play an important role in the recruitment of T cells by secreting chemokine ligands for CXCR3 and CXCR6. Finally, using the method we previously published[32], we performed migration assay of CD45RA$^+$ naïve CD8$^+$ T cells and CX3CR1$^{hi}$ effector CD8$^+$ T cells of PB from the arthritis-irAE patients (Fig. 4d; Supplementary Fig. 5). As expected, CD45RA$^+$ naïve CD8$^+$ T cells did not migrated well in response to CXCL9/10/11/16 (ratio relative to absence of CXCL9/10/11/16, mean ± SD, 1.04 ± 0.14, $P = 0.48$). In contrast, CX3CR1$^{hi}$ effector CD8$^+$ T cells migrated well in response to CXCL9/10/11/16 (ratio relative to absence of CXCL9/10/11/16, 1.90 ± 0.34, $P = 0.001$), suggesting the potential migration of CX3CR1$^{hi}$ effector CD8$^+$ T cells in PB into the joints mediated by CXCL9/10/11/16.

Taken together, clonotypes and trafficking analyses of T cells suggested that SF CXCR3$^{hi}$ CXCR6$^{hi/lo}$ effector CD8$^+$ T cells, recruited via CXCL9/10/11 and CXCL16 secreted by myeloid cells, may contribute to arthritis-irAE disease pathogenesis.

**Th17 cells are greatly enriched in SF from patients with arthritis-irAE after combined ICI therapy.** The effects of CTLA-4 inhibitors and PD-1 inhibitors on immune cells, especially on T cells, are distinct[33], and irAE profiles differ by ICI regimen[2]. In addition, recent studies revealed that clinical presentations of arthritis-irAE differ by ICI regimen[12,34]. Given these findings, we hypothesized that the mechanisms of arthritis-irAE could also differ by ICI regimen. To test our hypothesis, we divided our cohort into two groups based on ICI regimen and compared PB and SF immune profiles between the two groups. The first group comprised patients who developed arthritis after PD-1 inhibitor monotherapy (hereafter, PD-1 inhibitor arthritis) and the other group comprised patients who developed arthritis after combined ICI therapy (hereafter, combined ICI arthritis). Compared with the PD-1 inhibitor arthritis group, patients in the combined ICI arthritis group tended to develop arthritis later (time from first ICI infusion to the development of arthritis; mean ± SD; PD-1 inhibitor arthritis vs. combined ICI arthritis; 29.7 ± 20.2 weeks vs. 109.4 ± 126.3 weeks; $P = 0.05$); otherwise, in general, clinical parameters were similar between the two groups (Supplementary Data 1; Supplementary Table 2). We immunoprofiled PB and SF

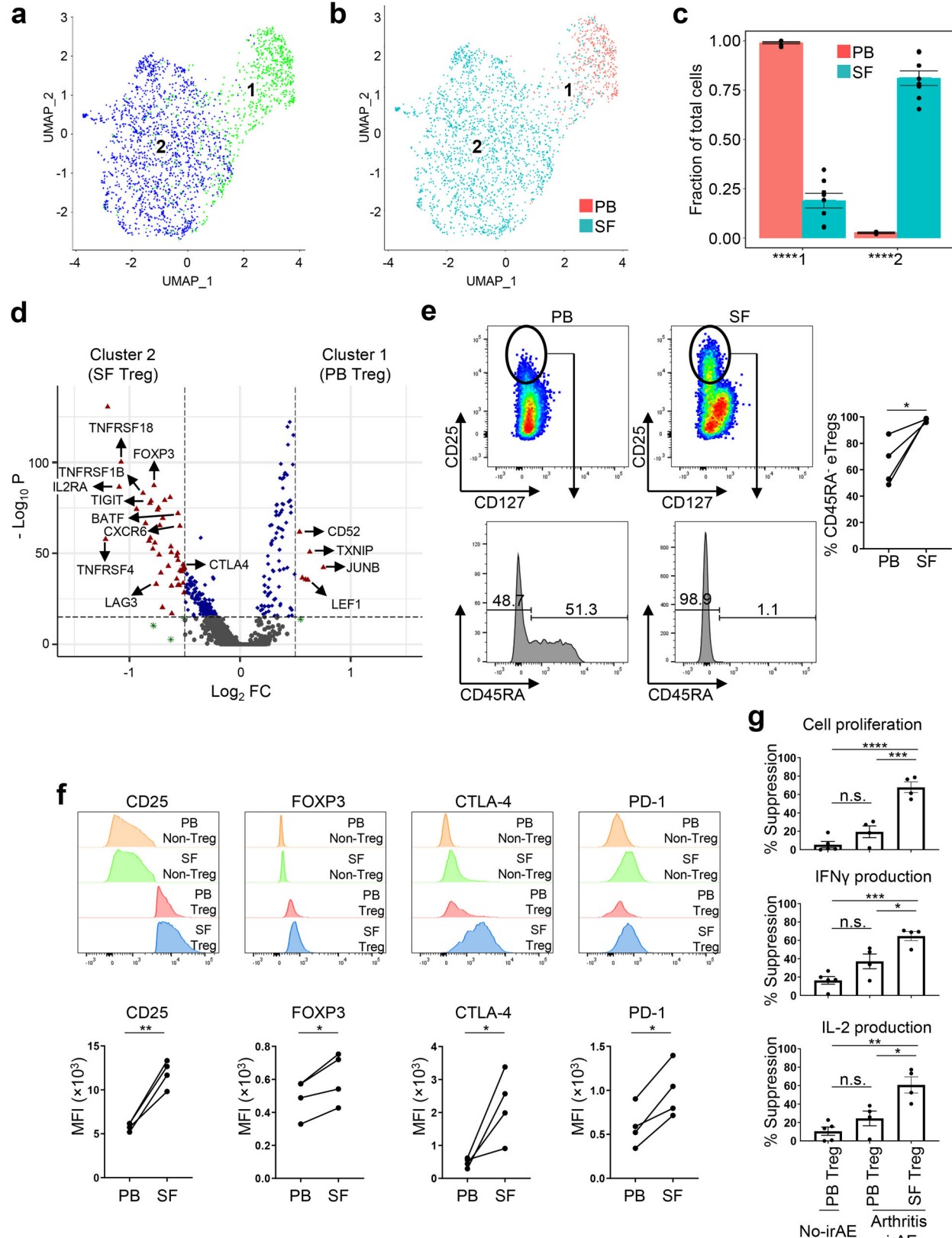

samples in these two groups, with special focus on T cells producing key effector cytokines, including IFNγ, IL-4, IL-17, and IL-21 (Fig. 5a–c). IL-17+ CD4+ T cells in SF were significantly enriched in the combined ICI arthritis group compared with the PD-1 inhibitor arthritis group (% within CD4+ T cells; PD-1 inhibitor arthritis vs. combined ICI arthritis; 1.52 ± 1.49 vs.

3.42 ± 0.89; P = 0.03). We further investigated the production of IFNγ and IL-17 in CD4+ T cells (Fig. 5a; Supplementary Fig. 6a). Proportions of IFNγ+ IL-17− CD4+ T (Th1) cells were similar between the two groups, whereas IFNγ− IL-17+ CD4+ T (Th17) cells were enriched in the combined ICI arthritis group (% within CD4+ T cells; PD-1 inhibitor arthritis vs. combined ICI arthritis;

**Fig. 3 Regulatory T cells (Tregs) are enriched and have enhanced suppressive functions in synovial fluid (SF) of patients with arthritis as an immune-related adverse event. a**, **b** Identification of two subclusters of Tregs and compartmental distribution of Treg subclusters. **c** Quantification of the subclusters and comparison of their cellular fractions between 8 peripheral blood (PB) and SF samples. Two-sided unpaired $t$ test. Subcluster 1, ****$P < 0.0001$; Subcluster 2, ****$P < 0.0001$. Bars indicate the mean and SEM. **d** Volcano plot showing differentially expressed genes between PB Tregs and SF Tregs. FC, fold change. See Supplementary Data 4 for all differentially expressed genes (DEGs) and Supplementary Data 5 for DEGs with $\log_2$ fold changes >0.5 or < −0.5 and $-\log_{10} P > 15$. **e** Representative flow cytometry plots of Tregs (left panels) and quantification analysis (right panel). $n = 4$ PB and matching SF samples. Two-sided paired $t$ test. *$P = 0.0354$. **f** Mean fluorescence intensity (MFI) of CD25, FOXP3, CTLA-4, and PD-1 on non-Tregs (CD25$^{lo}$ CD127$^{hi/lo}$) and Tregs (CD25$^{hi}$ CD127$^{lo}$). Representative flow cytometry plots (upper panels) and quantification analysis (lower panel) are shown. $n = 4$ PB and matching SF samples. Two-sided paired $t$ test. CD25, **$P = 0.0030$; FOXP3, *$P = 0.0224$; CTLA-4, *$P = 0.0460$; PD-1, *$P = 0.0116$. **g** Suppression of polyclonal response of naïve CD4$^+$ T cells with autologous PB Tregs or SF Tregs. The percentage of suppression of cell proliferation (CFSE dilution) and cytokine production were calculated as described in the Methods. Parallel experiments were performed with PB Tregs from patients in the no-irAE group. $n = 4$ PB and matching SF samples from arthritis-irAE patients; $n = 5$ PB samples from no-irAE patients. One-way analysis of variance. Cell proliferation, ***$P = 0.0002$, ****$P < 0.0001$; IFNγ production, *$P = 0.0201$, ***$P = 0.0003$; IL-2 production, *$P = 0.0130$, **$P = 0.0011$. Bars indicate the mean and SEM. IFNγ, interferon gamma; IL-2, interleukin-2. Source data are provided as a Source Data file.

$0.79 \pm 0.83$ vs. $1.89 \pm 0.64$; $P = 0.04$). Notably, IFNγ$^+$ IL-17$^+$ CD4$^+$ T cells, termed transient Th17 cells (t-Th17) and known to be enriched in SF from patients with juvenile idiopathic arthritis and rheumatoid arthritis, were also enriched in the combined ICI group (% within CD4$^+$ T cells; PD-1 inhibitor arthritis vs. combined ICI arthritis; $0.73 \pm 0.69$ vs. $1.54 \pm 0.30$; $P = 0.04$). Cytokine profiles in SF CD8$^+$ cells were similar, including enrichment of IL-17–producing CD8$^+$ T cells and IFNγ$^+$ IL-17$^+$ CD8$^+$ T (t-Tc17) cells; however, the difference did not reach statistical significance (% of IL-17$^+$ CD8$^+$ T cells; PD-1 inhibitor arthritis vs. combined ICI arthritis; $0.60 \pm 0.61$ vs. $1.37 \pm 0.82$; $P = 0.11$) (% of t-Tc17 cells; PD-1 inhibitor arthritis vs. combined ICI arthritis; $0.35 \pm 0.46$ vs. $0.97 \pm 0.66$; $P = 0.10$) (Fig. 5b; Supplementary Fig. 6b). In PB, key cytokine production profiles of CD4$^+$ and CD8$^+$ T cells were similar between the PD-1 inhibitor arthritis and combined ICI arthritis groups (Supplementary Fig. 6c, d).

Next, we compared the frequencies of Th1/Tc1, t-Th17/t-Tc17, and Th17/Tc17 cells in paired SF and PB samples from individual patients ($n = 4$ pairs in PD-1 inhibitor arthritis, $n = 4$ pairs in combined ICI arthritis) (Fig. 5c). Levels of Th1 cells were significantly increased in SF compared with matched PB in both PD-1 inhibitor arthritis and combined ICI arthritis groups (PD-1 inhibitor arthritis; SF Th1 vs. PB Th1; $46.8 \pm 12.5$ vs. $15.2 \pm 7.6\%$; $P = 0.03$) (Combined ICI arthritis; SF Th1 vs. PB Th1; $35.8 \pm 18.0$ vs. $5.2 \pm 1.8\%$; $P = 0.04$). In contrast, proportions of SF t-Th17 cells and SF Th17 cells were higher compared with PB t-Th17 cells and PB Th17 cells only in the combined ICI arthritis group (SF t-Th17 vs. PB t-Th17; $1.48 \pm 0.31$ vs. $0.29 \pm 0.33\%$; $P = 0.0015$) (SF Th17 vs. PB Th17; $1.86 \pm 0.74$ vs. $0.97 \pm 0.82\%$; $P = 0.04$). Frequencies of Tc1 cells were significantly higher in SF compared with PB in both the PD-1 inhibitor arthritis group and the combined ICI arthritis groups (PD-1 inhibitor arthritis; SF Tc1 vs. PB Tc1; $71.7 \pm 19.8$ vs. $26.8 \pm 13.5\%$; $P = 0.04$) (Combined ICI arthritis; SF Tc1 vs. PB Tc1; $63.2 \pm 22.4$ vs. $29.3 \pm 17.9\%$; $P = 0.0007$). Frequencies of t-Tc17 and Tc17 were in SF and PB were comparable in both the PD-1 inhibitor arthritis group and the combined ICI arthritis group. Finally, we measured soluble inflammatory cytokines in SF supernatant and serum (Supplementary Fig. 7). Levels of inflammatory cytokines, including IFNγ, IL-1β, IL-6, and IL-17A, in SF supernatant from arthritis-irAE groups (both the PD-1 inhibitor arthritis and combined ICI arthritis) was significantly higher compared with those from patients with osteoarthritis, suggesting that arthritis-irAE is inflammatory (Supplementary Fig. 7a). Levels of pro-inflammatory cytokines, including IL-6 and IL-17A, critical cytokines for Th17 cell differentiation and function[18], were higher in the combined ICI arthritis group than in the PD-1 inhibitor arthritis group, although the differences were not statistically significant (IL-6; PD-1 inhibitor arthritis vs. combined ICI arthritis; $2234 \pm 1,438$ vs. $2928 \pm 1684$ pg/mL; $P = 0.38$) (IL-17A: PD-

1 inhibitor arthritis vs. combination therapy arthritis; $14.5 \pm 16.8$ vs. $31.2 \pm 20.0$ pg/mL; $P = 0.25$). Although not reached statistical significances, the levels of inflammatory cytokines in serum of osteoarthritis patients were lower than those from no-irAE, PD-1 inhibitor arthritis, combined ICI arthritis groups (Supplementary Fig. 7b). Of note, serum concentrations of IL-6 and IL-17A were higher in the combined ICI arthritis group than in the PD-1 inhibitor arthritis group, but the differences were not statistically significant (Supplementary Fig. 7b).

Taken together, our immune profiling of SF and PB samples uncovered more pronounced Th17/Tc17 cell signatures in patients who developed arthritis after combined ICI therapy.

**Th17/Tc17 cells are steroid-resistant.** We next sought to determine whether distinct immune profiles between PD-1 inhibitor arthritis and combined ICI arthritis are associated with different arthritis-irAE clinical outcomes. We were particularly interested in steroid resistance in arthritis-irAE because steroids hinder antitumor immunity revived by the ICIs[7,8]. We prospectively observed our patients with arthritis-irAE for 12 months after SF and/or PB sample donation and monitored whether steroid monotherapy failed and DMARDs were warranted. Nine of 20 patients needed DMARDs; six patients needed biologic DMARDs (tocilizumab, $n = 4$; infliximab, $n = 1$; adalimumab, $n = 1$), two patients needed conventional DMARDs (methotrexate, $n = 1$; methotrexate + hydroxychloroquine, $n = 1$), and one patient needed combined biologic and conventional DMARDs (tocilizumab + sulfasalazine, $n = 1$). Previous studies showed that ICI regimen (combined ICIs) is a risk factor for steroid monotherapy failure in the treatment of arthritis-irAE[12,34]. Consistent with these studies, although not reached statistical significance ($P = 0.09$), we also observed that patients who developed arthritis after combined ICI therapy frequently experienced steroid monotherapy failure and required DMARDs (Fig. 6a). On the basis of our clinical-immunologic analyses (Figs. 5 and 6a), we hypothesized that Th17/Tc17 cells are steroid-resistant. To test this hypothesis, we analyzed the longitudinal PB samples from seven patients with arthritis-irAE who (1) were not receiving steroids at the time of the initial PB sample collection and (2) agreed to donate longitudinal PB samples (Fig. 6b–d). Demographic and clinical characteristics of the seven participants are shown in Table 2 and Supplementary Data 1. Five patients belonged to the PD-1 inhibitor arthritis group and two patients were in the combined ICI arthritis group. The second PB samples were collected at the median of 7 weeks after the first PB samples (range: 3–15 weeks). All patients were receiving steroid-based immune suppression when they donated the second PB sample; three patients were receiving steroid monotherapy and four were receiving DMARDs in addition to steroids (tocilizumab, $n = 2$; infliximab, $n = 1$; methotrexate, $n = 1$). The frequency of IFNγ$^+$

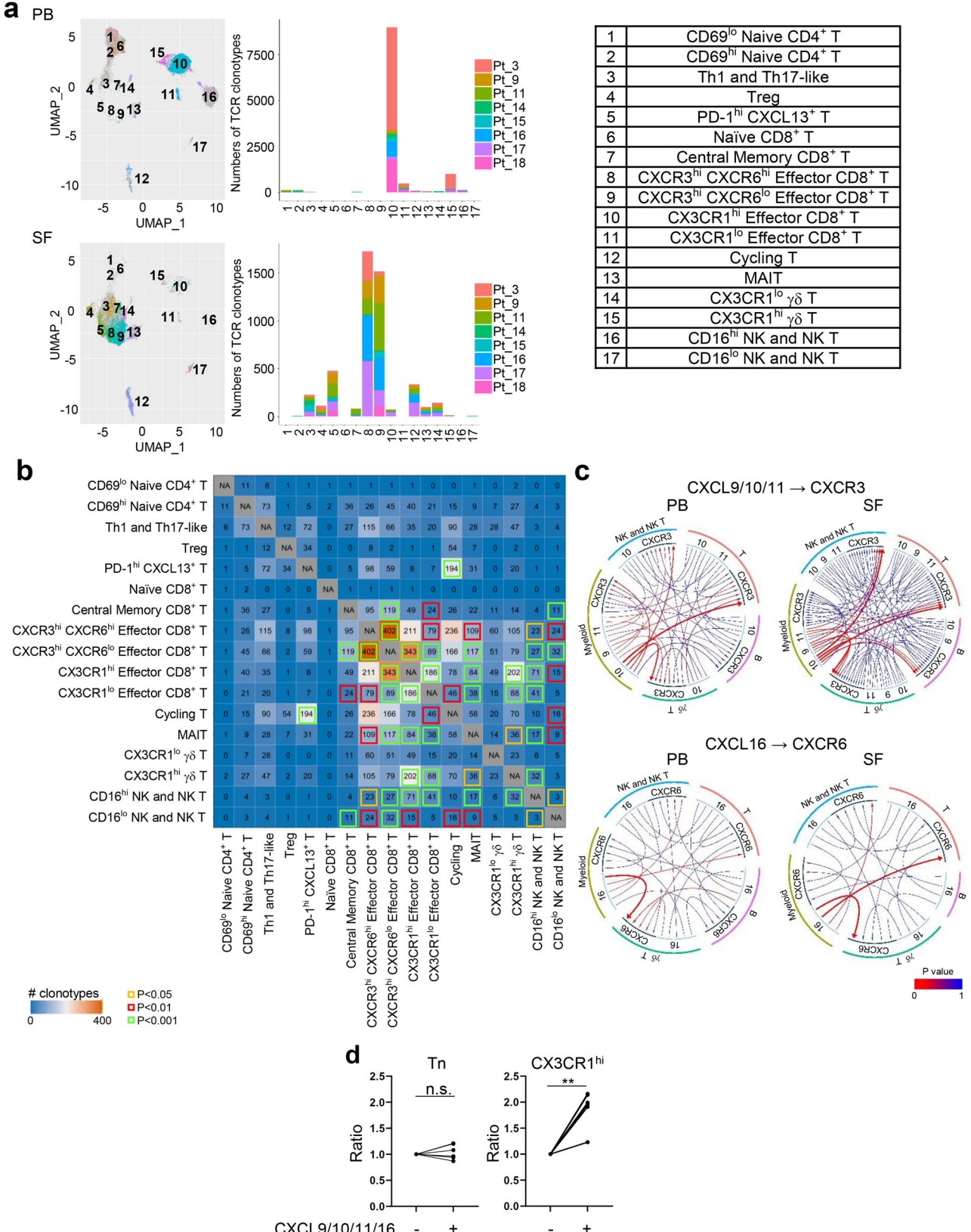

CD4+ T cells and Th1 cells was significantly decreased after treatment for arthritis-irAE, whereas the frequency of IL-17+ CD4+ T cells, t-Th17 cells, and Th17 cells was unchanged after treatment (Fig. 6b). Likewise, IFNγ+ CD8+ T cells and Tc1 cells were contracted after treatment of arthritis-irAE; however, the differences did not reach statistical significance (% of IFNγ+ CD8+ T cells; before

vs. after arthritis treatment; 32.9 ± 11.5 vs. 26.7 ± 15.6; $P = 0.07$) (% of Tc1 cells; before vs. after arthritis treatment; 32.5 ± 11.6 vs. 26.4 ± 15.7; $P = 0.07$) (Fig. 6c). Consistent with flow cytometry data, serum IFNγ was significantly reduced after the treatment of arthritis (Fig. 6d). Serum level of pro-inflammatory cytokines, TNFα and IL-6, likely secreted by myeloid cells[35,36], were also decreased after

**Fig. 4 CXCR3^hi CXCR6^hi/lo effector CD8+ T cells, recruited via CXCL9/10/11 and CXCL16 secreted by myeloid cells, may contribute pathogenesis of arthritis as an immune-related adverse event. a** UMAP view of the distribution of top 100 expanded T cell clones in natural killer (NK), NK T, and T cell clusters (left), number of top 100 expanded T cell clones across clusters from each patient (middle), and annotation of the 17 cell clusters (right). $n = 8$ PB and matching SF samples. PB, peripheral blood; SF, synovial fluid; TCR, T cell receptor; Treg, regulatory T cells; MAIT, mucosal-associated invariant T cells. **b** Heatmap showing T cell repertoire overlap across clusters. Numbers indicate the number of shared clonotypes between each cluster pair. Statistically significant overlap of T cell clones is indicated by colored boxes, based on a one-sided Fisher's Exact test followed by adjusting the false discovery rate using the Benjamini-Hochberg method. $n = 8$ PB and matching SF samples. **c** Circos plots showing interactions between major immune cells subsets via chemokine receptors (CXCR3 and CXCR6) and their ligands (CXCL9/10/11 and CXCL16). The lines were colored based on their statistically significance and the weight of the lines denotes their corresponding gene expression levels, the higher the expression level, the thicker the line. $n = 8$ PB and matching SF samples. **d** Ratio of migrated cells in response to CXCL9/10/11/16 to migrated cells in the absence of the chemokines. $n = 6$ PB samples from arthritis-irAE patients. Tn: naïve CD8+ T cells; CX3CR1^hi, CX3CR1^hi effector CD8+ T cells. Two-sided paired $t$ test. CX3CR1^hi, **$P = 0.0014$. Source data are provided as a Source Data file.

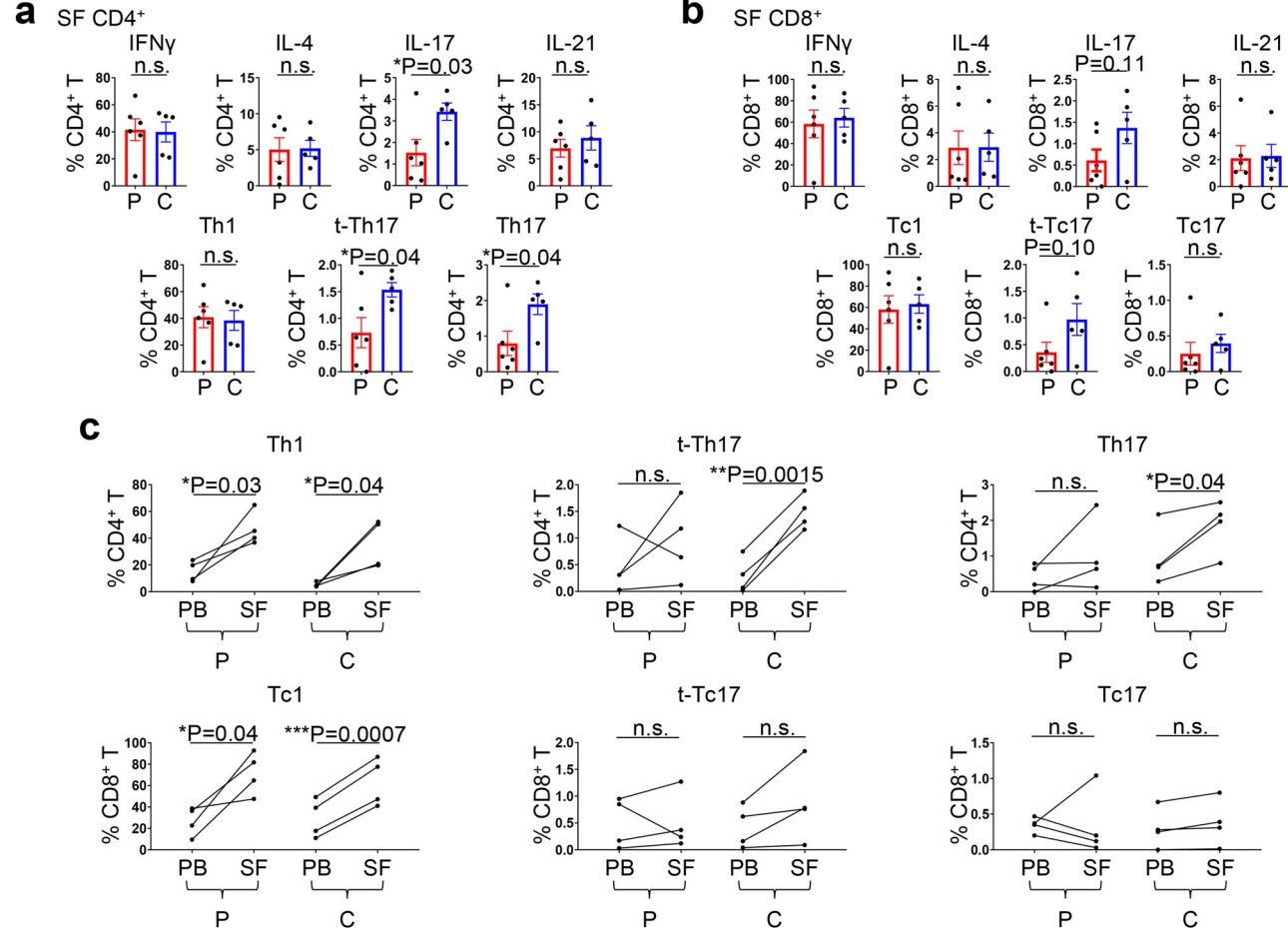

**Fig. 5 Th17 cells are enriched in the synovial fluid (SF) of patients who develop arthritis after combined immune checkpoint inhibitor (ICI) therapy.
a** Percentage of CD4+ T cells in SF producing key effector cytokines (upper panels) and percentage of Th1, transient (t-)Th17, and Th17 cells (lower panels). Two-sided unpaired $t$ test. *$P < 0.05$. Bars indicate the mean and SEM. IFNγ, interferon gamma; IL, interleukin. See Supplementary Table 2 for demographic and clinical profiles of patients who developed arthritis after PD-1 inhibitor monotherapy (PD-1 inhibitor arthritis; P in the figure) and patients who developed arthritis after combined CTLA-4 and PD-1 inhibitor therapy (combined ICI arthritis; C in the figure). $n = 6$ from the PD-1 arthritis group and 5 from the combined ICI arthritis group. **b** Percentage of CD8+ T cells in SF producing key effector cytokines (upper panels) and percentage of Tc1, t-Tc17, and Tc17 cells (lower panels). Two-sided unpaired $t$ test. Bars indicate the mean and SEM. $n = 6$ from the PD-1 arthritis group and 5 from the combined ICI arthritis group. **c** Frequencies of Th1/Tc1, t-Th17/t-Tc17, and Th17/Tc17 cells in 4 peripheral blood (PB) and matching SF samples. Two-sided paired $t$ test. Source data are provided as a Source Data file.

treatment. In contrast, serum levels of Th17 signature cytokines, including IL-17A, IL-17F, IL-21, IL-22, and IL-25[37], were constant.

Our data collectively suggested that Th17 and Tc17 cells may contribute to steroid resistance in arthritis-irAE, which is more associated with combined ICI arthritis.

**Discussion**

Despite clinical success, ICIs are associated with irAEs, including arthritis. In managing irAEs, including arthritis-irAE, establishing an optimal therapeutic strategy without impeding antitumor immunity is an unmet clinical need. To meet the need, it is

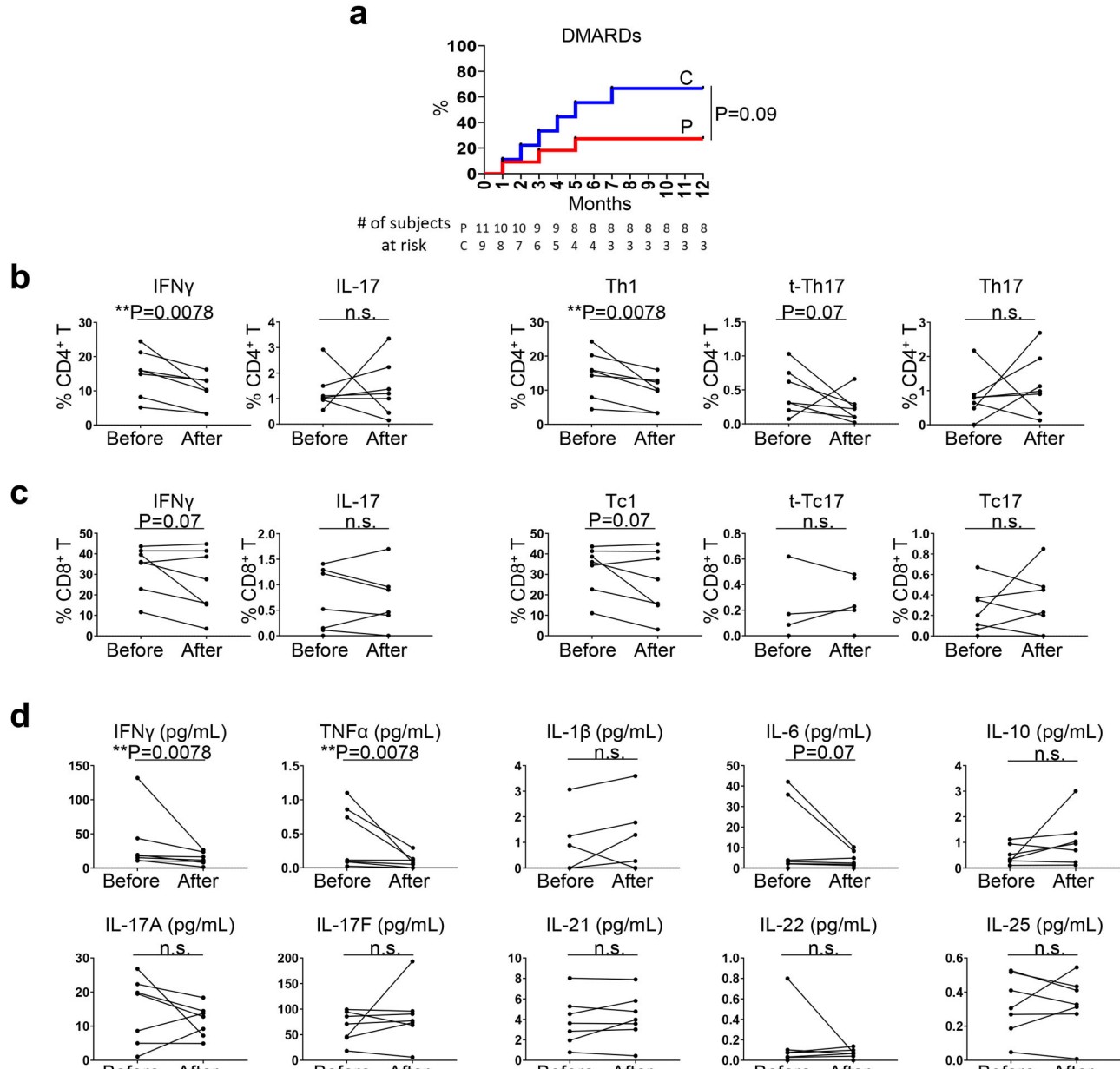

**Fig. 6 Th17 and Tc17 cells are steroid-resistant. a** Kaplan–Meier curves showing the proportion of patients whose steroid monotherapy failed and warranted disease-modifying anti-rheumatic drugs (DMARDs) over 12 months of follow-up based on ICI regimen. Two-sided Log rank test. P, PD-1 inhibitor arthritis; C, Combined ICI arthritis. **b** Percentage of interferon gamma (IFNγ)- and interleukin (IL)-17–producing CD4+ T cells in PB (left panels) and Th1, transient (t-)Th17, and Th17 cells (right panels) before and after treatment of arthritis from 7 patients with arthritis-irAE. One-sided Wilcoxon matched-pairs signed-rank test. **c** Percentage of IFNγ- and IL-17–producing CD8+ T cells in PB (left panels) and Tc1, t-Tc17, and Tc17 cells (right panels) before and after treatment of arthritis from 7 patients with arthritis-irAE. One-sided Wilcoxon matched-pairs signed-rank test. **d** Concentration of inflammatory cytokines in serum before and after treatment of arthritis from 7 patients with arthritis-irAE. One-sided Wilcoxon matched-pairs signed-rank test. Source data are provided as a Source Data file.

critical to elucidate the underlying mechanisms of arthritis-irAE. However, like other irAEs, there are no preclinical models fully recapitulating arthritis-irAE[9]. This underscores there is a fundamental need for comprehensive translational and clinical studies to have better understanding irAEs including arthritis-irAE[38]. Our clinical-molecular-immunologic analyses of a large number of samples from patients with arthritis-irAE revealed (1) a prominent IFNγ-producing T cells in SF; (2) expansion of SF Tregs with heightened suppressor functions; (3) potential role of effector CD8+ T cells; and (4) enhanced Th17 cell signatures in combined ICI arthritis with steroid resistance

(Fig. 7). We will discuss biological and clinical significances of each observation.

Interferon gamma (IFNγ)-producing T cells play a complex role in autoimmune arthritis[39]. At the early phase of the rheumatoid arthritis, Th1 cells facilitate the development of RA by activating macrophages as well as immunoglobulin class switching[40]. At later stages of rheumatoid arthritis, Th1 cells play a protective role by suppressing IL-17 production[41–43]. Of note, recent study showed the enrichment IFNγ+ CD8+ T cells in RA synovium[44]. Same study also revealed that RA synovial fibroblasts highly express IFNγ-inducible protein 30. In addition, the meta-analysis across

**Table 2 Demographic and clinical characteristics of patients included in the longitudinal peripheral blood analysis.**

| | |
|---|---|
| Median age (range), years | 57 (34–77) |
| Male, No. (%) | 2 (28) |
| ICIs, No. | |
| PD-1 inhibitor | 5 |
| Combined CTLA-4 and PD-1 inhibitors | 2 |
| Median time from first to second sample collection (range), weeks | 7 (3–15) |
| Arthritis treatment at the time of second blood sample collection, No. | |
| Steroid monotherapy | 3 |
| Steroid plus methotrexate | 1 |
| Steroid plus tocilizumab | 2 |
| Steroid plus infliximab | 1 |
| Median maximum dose of prednisone (or equivalent) between the first and second sample collections (range), mg | 20 (10–135) |

datasets of gene expression microarray demonstrated the pathogenic role of IFNγ in RA[45]. In our study, we observed that IFNγ producing T cells co-expressed multiple genes of inflammatory molecules (Fig. 2f) and SF Th1/Tc1 cells were enriched in SF compared to matching PB Th1/Tc1 cells in arthritis-irAE (Fig. 5c). Finally, Th1/Tc1 cell signatures and arthritis disease activity were diminished with steroid-based arthritis-irAE therapy (Fig. 6). Together, our data indicate that Th1/Tc1 cells contribute to the development of arthritis-irAE. As is seen in rheumatoid arthritis, Th1 cells might help macrophages and B cell isotype switch in arthritis-irAE, however, we do not exclude possibility that Th1 cells provide help to other immune cells, especially CD8+ T cells and neutrophils.

In this study, we observed that Tregs were enriched in SF. In addition, SF Tregs expressed more suppressor molecules and key transcription factors, including CTLA-4, CD25, and FOXP3, and SF Tregs showed superior suppressor functions over those observed in PB Tregs in in vitro suppressor assays (Fig. 3). Here, we raise three possibilities, which are not mutually exclusive. First, SF Treg functions might be enhanced to prevent excessive tissue damages. Second, since SF T cells in inflamed joints are thought to be more resistant to Treg suppression compared with matching peripheral blood (PB) T cells[46], SF Treg functions in arthritis-irAE were heightened accordingly. Finally, compositions of Treg subsets, which have their own target cells[47,48], might be different between SF and PB. Interestingly, studies analyzing SF and pairing PB samples from rheumatoid arthritis or juvenile idiopathic arthritis patients revealed that SF Tregs are more suppressive compared with matching PB Tregs[46,49]. It would be an important topic in the future to understand the biology of Tregs in arthritis-irAE versus classical autoimmune arthritis. In addition, recent study has showed that Treg functions might be inversely correlated with severity of irAEs, suggesting heterogeneity of Treg functions in irAEs[50]. In this regard, it would also be a critical future topic to investigate Treg biology between arthritis-irAE and non-arthritic irAEs. Finally, given that CTLA-4 is constitutively expressed, it would also be an important topic to compare Treg biology between PD-1 inhibitor arthritis-irAE and combined ICI arthritis-irAE.

Clonotypes and trafficking analyses of T cells suggested that clonally expanded CX3CR1hi effector CD8+ T cells in the blood migrate into joints, and our data strongly suggest that this process is mediated by chemokine receptors CXCR3 and CXCR6. CX3CR1hi CD8+ T cells in PB are known to be increased after PD-1 inhibitor treatment in patients with

melanoma and renal cell carcinoma[51,52], and CX3CR1hi CD8+ T cells play a critical role in tumor eradication in vivo[52]. Therefore, it is plausible that CX3CR1hi effector CD8+ cells from tumors might cross reacted to tissues (in this study, joint), causing subsequent inflammation. This hypothesis is partially supported by previous studies showing overlaps of TCR between tumors and inflamed cardiac muscles in patients with myocarditis-irAE, as well as overlaps of TCR between tumors and inflamed lungs in patients with pneumonitis-irAE[53,54]. However, it should also be noted that CX3CR1hi T cells, producing IFNγ, TNFα, granzyme A, and perforin, were increased in PB of rheumatoid arthritis patient[55]. Therefore, although not mutually exclusive, it is possible that cancer patients who are genetically predisposed to autoimmune arthritis might develop arthritis-irAEs after they lose immune tolerances with ICI therapy. It would be interesting to determine whether circulating CX3CR1hi effector CD8+ T cells are specific to tumor-antigens and/or self-antigens in future studies. Murine and translational studies suggested that CXCR3 and CXCR6 are key elements for effector T cells to migrate into the joints in rheumatoid arthritis, and MDX-1100, anti-CXCL10 monoclonal antibody, showed clinical efficacy in rheumatoid arthritis[56,57]. Similar to rheumatoid arthritis, in arthritis-irAE, SF T cells highly expressed CXCR3 and CXCR6, and their matching ligands, CXCL9/10/11, and CXCL16 were mainly produced by myeloid cells (Fig. 4c; Supplementary Figs. 3c, 4). Nevertheless, we do not exclude the role of non-hematopoietic cells as the source of these chemokine ligands. Indeed, human and murine studies showed that fibroblast-like synoviocytes produce CXCL16 in rheumatoid arthritis[57,58]. Notably, CXCL9/10 and CXCR3 are critical for anti-tumor effector T cells to migrate into tumors[59], and a recent study also showed that CXCR6 also play a critical role in anti-tumor immunity in hepatocellular carcinoma[60]. It would be an interesting and important topic in irAE research to investigate mechanisms by which effector T cells migrate into tumor and inflamed organs.

Patients with combined ICI arthritis had enhanced Th17 cell signatures in both SF and PB (Fig. 5; Supplementary Figs. 6–7). Furthermore, we observed that combined ICI arthritis tended to be resistant to steroid monotherapy and required a DMARD more often than PD-1 inhibitor arthritis (Fig. 6a). Studies in both humans and mice have demonstrated that Th17 cells are resistant to steroid therapy[61,62], and consistently, we observed that t-Th17 cells or Th17 cells were constant whereas Th1 cells were contracted after steroid-based therapy for arthritis-irAE (Fig. 6). Of note, CTLA-4 inhibitors enhance Th17 differentiation both in humans and mice[63,64]. Studies revealed that Th17 cells are plastic and could be converted into t-Th17 and non-classical Th1 (ex-Th17) cells in response to IL-12[65]. Importantly, t-Th17 cells and non-classical Th1 cells are inflammatory and their accumulation in the synovium was observed in rheumatoid arthritis and juvenile idiopathic arthritis[65–67]. Taken together, we speculate that in combined ICI arthritis, steroid-resistant Th17 cells, differentiated by CTLA-4 inhibitor therapy, might undergo plasticity into Th1 cells, which in turn help CD8+ T cells in arthritis-irAE.

Some differences and similarities in the molecular and immune landscapes of arthritis-irAE, classic autoimmune arthritis, and non-arthritic-irAEs seem to be apparent. A recent study analyzing the synovium of rheumatoid arthritis showed that granzyme K+ granzyme B+ GNLY+ CD8+ cells and PD-1hi CXCL13hi peripheral helper T cells were expanded[44]. Similarly, we observed these cell populations in the SF of patients with arthritis-irAE (Fig. 4). In contrast, IL-1β+ pro-inflammatory monocytes were expanded in rheumatoid arthritis synovia, whereas CD14hi CD16hi neutrophils

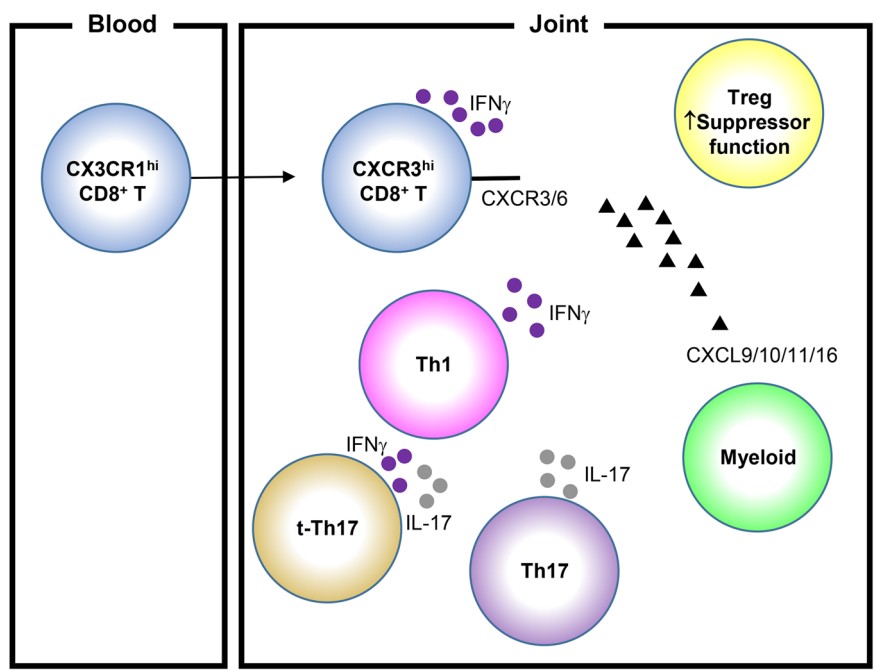

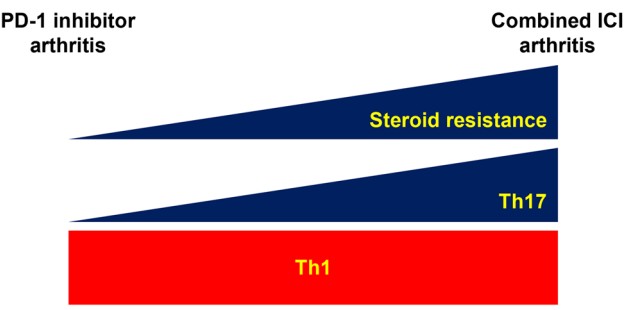

**Fig. 7 Graphical abstract of the study.** ICI immune checkpoint inhibitors; IFNγ interferon gamma.

were expanded in arthritis-irAE (Fig. 1). In addition, although B cells, especially antibody-producing plasmablasts, were expanded in rheumatoid arthritis synovia, SF B cells were scarce in arthritis-irAE. Our observation is supported by a recent report showing lack of B cells on synovial tissues of arthritis-irAE[68]. Nevertheless, we cannot entirely ignore B cells in the pathogenesis of arthritis-irAE given their interactions with other immune cell subsets via chemokines and chemokine receptors (Fig. 4c). Another study of SF and autologous PB samples from patients with psoriatic arthritis showed that HLA-DR[hi] CD8[+] T cells were clonally expanded in SF[69]. The transcriptomic landscape of these HLA-DR[hi] CD8[+] T cells was similar to that of CXCR3[hi] CXCR6[lo] effector CD8[+] cells in SF in our cohort (Fig. 2; Supplementary Data 3). However, tissue resident ZLF683[+] CD8[+] cells, which are enriched in the joints of patients with psoriatic arthritis, were not observed in our cohort. A recent study analyzing patients with colitis-irAE provided an opportunity to compare the molecular and immune profiles of colitis-irAE and arthritis-irAE[70]. As in arthritis-irAE, cytotoxic effector CD8[+] T cells, effector Treg cells, and cycling T cells were expanded in colitis-irAE colon tissues. In addition, IFNγ is a key mediator in colitis-irAE, as in arthritis-irAE. As in arthritis-irAE, CXCR3 and CXCR6 were suggested to play a key role in recruiting T cells into colitis-irAE sites. It is noteworthy that 35% of patients (7/20) had colitis-irAE prior to the arthritis-irAE (Table 1;

Supplementary Data 1). Nevertheless, we did not observe IgA-producing plasmablasts in arthritis-irAE joints. Taken together, unique mechanisms are present among arthritis-irAE, non-arthritic irAEs, and classical autoimmune arthritis in terms of T cell functional status, T cell-B cell interactions, and the role of B cells and myeloid cells.

The current study has some limitations. Although we analyzed samples using integrative approaches, because arthritis-irAE is a relatively rare event[6], the sample size were small and our cohort was heterogeneous. Second, we did not analyze tumor samples. Addressing the question of whether arthritis-irAE is a tumor-dependent response would be an important topic for future studies. Likewise, investigations of the antigens and their sources would be a critical topic in future researches. Recent study showed frequency of nivolumab-bound Ki67[+] CD8[+] T cells in PB from non-small cell lung cancer patients whose tumor responded well to nivolumab[71]. It would also be interesting to characterize ICI-bound T cells in tumor, PB, and SF from patients with arthritis-irAE and such characterization might provide us insights on pathogenesis of arthritis-irAE. Third, although we were indirectly able to compare the immune landscape of arthritis-irAE with that of classic autoimmune arthritis or non-arthritic irAEs, a head-to-head comparison would have provided us stronger insights into the mechanisms underlying arthritis-

irAE. Fourth, eight patients (40%) had received systemic steroids within 4 weeks prior to the sample collections, and two patients (10%) were receiving systemic steroids at the time of the sample collections. Exposure to steroids might have altered the immune profiles. Fifth, clinical data related to tumors (primary vs. recurrent; cancer stages; duration from the diagnosis of the cancer to the ICI therapy; and cancer therapy prior to the ICI treatment) as well as tumor outcomes after ICI therapy were missed in this study. Further studies with comprehensive samples of PB, inflamed tissues, and tumors from a substantial number of patients with arthritis-irAE for comparison with classic autoimmune arthritis and non-arthritic irAEs will be necessary to further understand the pathogenesis of arthritis-irAE. In parallel, to overcome the challenge and determine cellular/molecular mechanisms underlying arthritis-irAE, it would be essential to develop and utilize arthritis-irAE murine models.

In conclusion, our holistic analyses of patients with arthritis-irAE unmasked molecular and immunologic landscapes of arthritis-irAE. Our results will pave the way for understanding the mechanisms of arthritis-irAE, as well as identifying biomarkers to predict development and steroid-resistance of arthritis-irAE and therapeutic targets while preserving antitumor immunity.

## Methods

**Patients and collection of tissue samples**. This study was approved by the institutional review board at The University of Texas MD Anderson Cancer Center (IRB No: PA16-0935). Prior to the procedures (diagnostic arthrocentesis and/or venipuncture), participants provided written informed consent, allowing collection of residual SF and/or PB samples as well as prospective follow-up for 12 months. We collected residual SF and/or PB from 20 patients who newly developed arthritis after ICI therapy. As serum-negative controls, we collected PB samples from patients who had not developed irAEs at least 12 weeks after initiating ICI therapy. For SF supernatant-negative controls, we collected SF samples from patients with osteoarthritis. The patients met the American College of Rheumatology diagnostic criteria for osteoarthritis[10]. Participants included both male and female patients ranging in age from 34 to 77 years. Detailed information about individual participants can be found in Supplementary Data 1.

**Clinical assessment**. Information on age, sex, body mass index, tumors, pattern of arthritis, history of irAEs prior to the arthritis, onset of the arthritis, CDAI, erythrocyte sedimentation rate, C-reactive protein, anti-nuclear antibody, rheumatoid factor, and anti-cyclic citrullinated peptide antibody were obtained from the medical record. We followed patients with arthritis-irAE for 12 months after the sample collection. At the end of the follow-up, we noted the number of patients who required DMARDs. We also calculated the accumulated dose of prednisone (or equivalent) over 12 months.

**Isolation of cells**. SF samples were incubated with 10 IU hyaluronidase (Sigma-Aldrich) at 37 °C for 15 min. After incubation, the sample was centrifuged at 500 $g$ for 10 min, and SF supernatant was obtained. The cell pellet was washed with 1× phosphate-buffered saline solution (GIBCO), and SF cells were cryopreserved in a mixture of 90% bovine serum albumin and 10% dimethyl sulfoxide (Sigma–Aldrich). For PB samples, the plasma was isolated after centrifugation at 500 $g$ for 10 min. Peripheral blood mononuclear cells (PBMCs) were isolated with the FiColl gradient technique (Sigma–Aldrich) and cryopreserved like SF cells.

**Sample processing for scRNAseq and scTCRseq**. Cryopreserved SF cells and PBMCs were thawed, washed, and resuspended with 1× phosphate-buffered saline containing 0.05% RNase-free bovine serum albumin (ThermoFisher Scientific) at a final volume of <35 μL. Cell counts and viabilities were determined using trypan blue exclusion and counted on a Countess II FL automated cell counter (Life Technologies). Reagents, consumables, reaction master mixes, reaction volumes, cycling numbers, cycling conditions, and clean-up steps were completed following 10X Genomics' guidelines (found in 10X Genomics' Chromium Single Cell V(D)J Reagent Kits User Guide-Revision L). cDNA was allocated for preparation of a gene expression library (Chromium Single Cell 5' Library Construction Kit; Cat # 1000020) or TCR enrichment/library preparation (Chromium Single Cell V(D)J Enrichment Kit, Human T cell; Cat # 1000005). Quality control steps after cDNA amplification and library preparation steps were carried out by running Qubit[TM] 1X dsDNA HS Assay (Thermo Fisher Scientific) along with Bioanalyzer High-Sensitivity DNA Analysis (Agilent) for concentration and quality assessments. Library sample concentrations were verified with quantitative polymerase chain reaction using a KAPA Library Quantification Kit (Roche). Samples were

normalized to 10 nM for pooling. The gene expression libraries and TCR libraries were pooled in a ratio of 5 volumes gene expression library to 1 volume TCR library. The pool was sequenced using a NovaSeq6000 S4-200 cycle flow cell (Illumina). The run parameters used were 26 cycles for read 1, 91 cycles for read 2, 8 cycles for index 1, and 0 cycles for index 2, as stipulated in the protocol mentioned above. Raw sequencing data (fastq file) was demultiplexed and analyzed with 10X Genomics Cell Ranger software using the standard default settings and the cell ranger count command to generate html quality control metrics and cloupe/vloupe files for each sample.

**scRNAseq data processing, filtering and batch effect correction**. Raw single-cell RNA-seq data were pre-processed using 10X Genomics Cell Ranger (v3.1.0), including demultiplexing cellular barcodes, aligning sequencing reads to the human reference genome (GRCh38, v3.0.0, from 10X Genomics), and generation of the cell by gene expression matrix. Detailed quality control (QC) metrics were generated and evaluated, samples and cells were carefully and rigorously filtered to obtain high-quality data for downstream analyses[72]. Briefly, for quality filtering, cells with low complexity libraries (in which detected transcripts are aligned to <200 genes) were filtered out and excluded from subsequent analyses. This filter will help remove cell debris, empty drops, and low-quality cells. Likely dying or apoptotic cells where >15% of transcripts derived from the mitochondrial genome were also excluded. Next, likely doublets or multiplets were detected and carefully removed through a multi-step approach as described in our recent studies[73–75]. Doublets and multiplets were identified by the following methods: (1) library complexity: cells with high complexity libraries (in which detected transcripts are aligned to >6500 genes) were removed; (2) Cluster distribution: doublets or multiplets likely form distinct clusters with hybrid expression features and exhibit an aberrantly high gene count; (3) cluster marker gene expression: cells of a cluster express markers from distinct lineages (e.g., cells in the T-cell cluster co-expressed myeloid cell markers and vice versa); (4) doublet detection algorithm: Scrublet[76], an algorithm to predict doublets in scRNA-seq data, was applied to further identify and clean doublets that could have been missed by steps 1–3. We carefully reviewed canonical marker genes expression on UMAP plots and repeated the above steps multiple times to ensure elimination of most barcodes associated with cell doublets. After doublets removal, a total of 116,797 cells were retained for downstream analyses. Lastly, we employed Harmony[77], one of the top-ranked methods for batch effect correction to iteratively remove batch effects present in the PCA space.

**Unsupervised clustering and subclustering analysis**. Seurat v3 (version 3.2.2)[78] was applied to the normalized gene-cell matrix to identify highly variable genes (HVGs) for unsupervised cell clustering. To identify HVGs, the *vst* method in the Seurat package was run with default parameters. Principal component analysis (PCA) was performed on the top 2,000 HVGs. The elbow plot was generated with the *ElbowPlot* function of Seurat and based on which, the number of significant principal components (PCs) was determined. The *FindNeighbors* function of Seurat was used to construct the Shared Nearest Neighbor (SNN) Graph, based on unsupervised clustering performed with Seurat function *FindClusters*. Different resolution parameters for unsupervised clustering were then examined in order to determine the optimal number of clusters. For visualization, the dimensionality was further reduced using Uniform Manifold Approximation and Projection (UMAP) method[79] with Seurat function *RunUMAP*. The number of PCs used to calculate the embedding was the same as that used for clustering. Three rounds of clustering and subclustering analyses were performed to identify major immune cell clusters and subcluaters. In the first round, 200-nearest neighbors of each cell were determined based on 40 PCs to construct SNN graph. The clustering was performed with resolution 0.6 and each cluster was annotated by known markers (see Determination of major cell types and cell states for details). The second and third rounds of subclustering were performed on NK/NK T/T cell clusters and Treg cell cluster, respectively. For NK, NK T, and T cell subclustering, 30-nearest neighbors of each cell were determined based on 30 PCs to construct SNN graph. The clustering was performed with resolution 0.8. For Treg subclustering, the SNN graph was constructed based on 30-nearest neighbors of each cell that were determined by 30 PCs. The clustering was performed with resolution 0.1.

**Determination of major cell types and cell states**. We determined the major immune cell subsets using a similar approach as previously described[80,81]. First, differentially expressed genes (DEGs) of each cluster were identified using the *FindMarkers* function in Seurat. We carefully reviewed top 50 DEGs for each clusters with special focus on well-studied canonical markers including CD3D/CD3E (T cells), CD3D/CD3E/TCF7 (naïve T cells), CD3D/CD3E/CD4 (CD4+ T cells), CD3D/CD3E/CD8A/CD8B (CD8+ T cells), CD3D/CD3E/CD8A/CD8B/KLRG1 (effector CD8+ T cells), CD3D/CD3E/TRDC/TRDV (γδ T cells), NCAM in the absence of CD3D/CD3E (NK cells), CD3D/CD3E/NCAM (NK T cells), CD19 (B cells), CD19/CD27/CD38/SDC1 (plasmablasts), CD14 (monocytes/macrophages), FCGR3A (neutrophils), CD1c/HLA-DR (dendritic cells), and PPBP (megakaryocytes). Multiple layers of information including the cluster distribution, cluster specific genes in particularly the top 50 DEGs, canonical cell lineage

markers were integrated and carefully reviewed to define cell types and cell transcriptomic states.

**scTCRseq data analysis**. T cell receptor (TCR) repertoire analysis was performed using a similar approach as we previously described[75]. Briefly, raw scTCRseq data were processed using Cell Ranger (v3.1.0) from 10X Genomics. The FASTQ reads were aligned to human GRCh38 V(D)J reference genome using the *cellranger vdj* function. The CDR3 motifs were located, followed by paired clonotype calling and annotation. The clonal fraction of each identified clonotype was then calculated. The productivity was determined for each clonotype and only productive clonotypes were included in subsequent analysis. On average, 69.0% of CD3+ T cells were associated with at least one productive TRA or TRB rearrangements. Of all detected clonotypes, 50.6% contained one paired TRA and TRB sequences. Only 2.7% of clonotypes contained more than one TRA or TRB sequences. The landscape, diversity, and clonal expansion of TCR repertoire was then assessed and repertoire overlap analysis was carried out based on the similarity of the paired TCRAV/TCRBV segments and the CDR3 sequences. The TCR clonotype data was then integrated with the T-cell phenotype data inferred from single cell gene expression analysis based on the shared cell barcodes.

**Cell-to-cell interaction analysis**. To identify significant ligand-receptor pairs between myeloid and lymphoid cells, first, we defined the highly expressed genes using a similar approach as described previously[82]. A ligand or receptor is considered as "expressed" in a certain cell type if >20% cells had a expression level log2 (normalized UMI + 1)>0. Subsequently, we matched these highly expressed genes with the curated list of ligand-receptor pairs from iTALK[83] to identify the potential ligands and receptors, focused on chemokines and chemokine receptors. We assume two cell types have interaction when one highly expressed ligand/receptor and the other one highly expressed the paired receptor/ligand. For the differential interactome analysis, we found the differentially expressed genes in each cell type using the function *FindMarkers* from Seurat with the parameter min.pct set to 0.2 and then matched these DEGs with the curated list of ligand-receptor pairs from iTALK. We also scored each interaction using the same way as herein described[82].

**Flow cytometry**. Cryopreserved SF cells and PBMCs were thawed, washed, and stained with flow cytometry antibodies. For cytokine intracellular staining, SF cells and PBMCs were stimulated with cell activation cocktail (Biolegend, Cat No: 423303) containing phorbol 12-myristate 13-acetate, ionomycin, and brefeldin for 4 hours. Cells were stained for surface molecules, fixed with BD Fixation/Permeabilization solution (BD CytoFix/CytoPerm™, Cat No: 51-2090KZ), permeabilized with BD PERM/Wash™ buffer (BD PERM/ Wash™ solution, Cat No: 51-2091KZ), and stained with antibodies to IFNγ, IL-4, IL-10, IL-17A, and IL-21□ For FoxP3 and Ki67 staining, cells were stained for surface molecules, fixed, and permeabilized with eBioscience™ FoxP3/Transcription staining buffer set (Cat No: 00-5523-00). Subsequently, the cells were stained for FoxP3 and Ki67. Stained samples were acquired using LSR II FORTESSA X-20 (BD Biosciences) and BD FACS DIVA (version 8.0.1; http://bdbiosciences.com), and analyzed with FlowJo software (TreeStar). Following antibodies were used for the flow cytometry; Anti-Human CD16 BUV 395 (BD Biosciences, Cat No: 563785; Dilution 1:20), Anti-Human CD27 BUV 395 (BD Biosciences, Cat No: 563815; Dilution 1:50), Anti-Human CD4 BUV 395 (BD Biosciences, Cat No: 563550; Dilution 1:50), Anti-Human CD56 Brilliant Violet 421 (Biolegend, Cat No: 362552; Dilution 1:50), Anti-Human γδ TCR Brilliant Violet 421 (Biolegend, Cat No: 331218; Dilution 1:50), Anti-Human IL-4 Brilliant Violet 421 (Biolegend, Cat No: 500826; Dilution 1:50), Anti-Human CCR7 Brilliant Violet 421 (Biolegend, Cat No: 353208; Dilution 1:50), Anti-Human Ki67 Brilliant Violet 421 (Biolegend, Cat No: 652411; Dilution 1:50), Anti-Human CD19 Brilliant Violet 785 (Biolegend, Cat No: 302240; Dilution 1:50), Anti-Human CD45RA Brilliant Violet 785 (Biolegend, Cat No: 304140; Dilution 1:50), Anti-Human PD-1 Brilliant Violet 785 (Biolegend, Cat No: 367432; Dilution 1:50), Anti-Human CD3 PerCP (Biolegend, Cat No: 344808; Dilution 1:50), Anti-Human HLA-DR FITC (Biolegend, Cat No: 307620; Dilution 1:50), Anti-Human CD56 FITC (Biolegend, Cat No: 318304; Dilution 1:50), Anti-Human FoxP3 FITC (Biolegend, Cat No: 320106; Dilution 1:25), Anti-Human CD123 PE (Biolegend, Cat No: 306006; Dilution 1:50), Anti-Human CD19 PE (Biolegend, Cat No: 302208; Dilution 1:50), Anti-Human IL-21 PE (BD Biosciences, Cat No: 562042; Dilution 1:20), Anti-Human CD25 PE (Biolegend, Cat No: 356104; Dilution 1:50), Anti-Human CX3CR1 PE (Biolegend, Cat No: 355704; Dilution 1:50), Anti-Human CD24 PE-Dazzle (Biolegend, Cat No: 311134; Dilution 1:50), Anti-Human CTLA-4 PE-Dazzle (Biolegend, Cat No: 369616; Dilution 1:25), Anti-Human IFNγ PE-Dazzle (Biolegend, Cat No: 502546; Dilution 1:25), Anti-Human CD11c PE-Cy7 (Biolegend, Cat No: 337216; Dilution 1:20), Anti-Human CCR7 PE-Cy7 (Biolegend, Cat No: 353226; Dilution 1:50), Anti-Human CD25 PE-Cy7 (Biolegend, Cat No: 302612; Dilution 1:50), Anti-Human IL-17A PE-Cy7 (Biolegend, Cat No: 512315; Dilution 1:40), Anti-Human CD4 PE-Cy7 (Biolegend, Cat No: 300512; Dilution 1:50), Anti-Human CD4 APC (Biolegend, Cat No: 300514; Dilution 1:50), Anti-Human IL-10 APC (Biolegend, Cat No: 506806; Dilution 1:40), Anti-Human IL-2 APC (Biolegend, Cat No: 500310;

Dilution 1:25), Anti-Human CD14 Alexa Fluor 700 (Biolegend, Cat No: 301822; Dilution 1:50), Anti-Human CD8 Alexa Fluor 700 (Biolegend, Cat No: 300920; Dilution 1:50), Anti-Human CD127 Alexa Fluor 700 (Biolegend, Cat No: 351344; Dilution 1:50), Anti-Human CD45 APC-Fire (Biolegend, Cat No: 368518; Dilution 1:50), and LIVE/DEAD Zombie Aqua™ (BioLegend).

**In vitro Treg suppressor assay**. Live CD45+ CD3+ CD4+ CCR7+ CD45RA+ naïve CD4+ T cells (responder T) and live CD45+ CD3+ CD4+ CD25hi CD127lo Tregs from PB samples from patients with arthritis-irAE were sorted with a BD FACS ARIA. In parallel, autologous SF Tregs were sorted. As a control, we sorted responder T cells and Tregs from PB samples from patients in the no irAEs group. Responder T cells were resuspended at a concentration of $10 \times 10^7$ cells/mL and incubated with CFSE (5 nM) (Biolegend, Cat No: 423801) for 20 minutes at 37 °C. CFSE-labeled responder T cells (10,000 cells/well) were cultured with CD3/CD28 Dynabeads (beads-to-cell ratio = 1:25) in the absence of Tregs or in the presence of either PB or SF Tregs (20,000 cells/well) for 4 days. Two hours before harvest, brefeldin A (Biolegend) was added at a ratio of 1:1000. Harvested cells were fixed, permeabilized, and stained with IFNγ PE-Dazzle and IL-2 APC. Percentages of proliferation (CFSE-negative) and cytokine production (IFNγ and IL-2) in responder T cells were measured. The percentage of suppression was calculated using the published method:[84]

$$100 - \frac{\% \text{ of proliferating (or cytokine} - \text{secreting) cells in the presence of Treg}}{\% \text{ of proliferating (or cytokine} - \text{secreting) cells in the absence of Treg}} \times 100$$

**Migration assay**. Migration assay was performed as we previously published[32], In brief, we sorted naïve CD8+ T cells (CD3+ CD8+ CD45RA+ CX3CR1lo) and CX3CR1hi effector CD8+ T cells (CD3+ CD8+ CD45RA− CX3CR1hi) from PB samples of patients with arthritis-irAE using FACs Aria. Sorted cells were washed and resuspended in migration medium (RPMI 1640 with 0.5% fetal bovine albumin) at $50 \times 10^3$ cells/mL. A total of $25 \times 10^3$ cells were loaded in a 8-mm pore transwell (BD Falcon, Cat#: Costar 3464). In the lower chamber, 500 μL of migration medium was placed in the presence/absence of 1 μg/mL of CXCL9 (R&D Systems, Cat #: 392-MG), CXCL10 (R&D Systems, Cat #: 266-IP), CXCL11 (R&D Systems, Cat #: 672-IT), and CXCL16 (R&D Systems, Cat #: 976-CX). Cells were allowed to migrate at 37 ˚C for 3 h. Subsequently, the cells in the lower chamber were collected and loaded for flow cytometry. Just before samples were loaded, $20 \times 10^3$ of beads (Spherotech) were added into each tube to standardize cell numbers between tubes. Events were analyzed with FlowJo software to calculate the ratio of migrated cells in response to the chemokines to migrated cells in absence of chemokines.

**HLA-B27 typing**. DNA was isolated from cryopreserved SF or PB samples using QIAsymphony kits (Qiagen, Redwood City, CA). HLA-B low-resolution typing was performed by Sequence-Specific Oligonucleotide method using LABType typing kit (One Lambda, Canoga Park, CA). The HLA-B27 positivity was determined by the presence of the HLA-B27 allele.

**Cytokine measurement**. Levels of IFNγ, TNFα, IL-1β, IL-6, IL-10, IL-17A, IL-17F, IL-21, IL-22, and IL-25 in SF supernatant and serum were measured by multiplex using commercially available kits (U-Plex Th17 Combo 2; Meso Scale Discovery) according to the manufacturer's instructions.

**Statistical analysis**. The mean differences in percentages of immune cells and concentrations of cytokines were determined by unpaired *t* test for two groups (Figs. 1e, 2d, 3c, 5a–b; Supplementary Figs. 6c–d) or one-way analysis of variance for three or more than three groups (Figs. 2f, 3g; Supplementary Figs. 3b, 7a–b) with the Bonferoni correction. For TCR overlap analyses (Fig. 4b), we used a one-sided Fisher's Exact test followed by adjusting the false discovery rate using the Benjamini-Hochberg method. For ligand-receptor pair analyses (Fig. 4c), we used an identical methodology described previously[82]. The mean differences in percentage of immune cells between tissues (SF compared with autologous PB) were compared using a paired *t* test (Figs. 1f, 3e–f, 5c; Supplementary Fig. 1f). Proportional differences in the requirement of DMARDs at 12-month follow-up by ICI regimen were examined using the log-rank test (Fig. 6a). The difference in proportions of T cell subsets, as well as mean serum cytokines before and after treatment of arthritis, was determined by the Wilcoxon matched-pairs signed-rank test (Fig. 6b–d). $P < 0.05$ were considered statistically significant.

**Reporting summary**. Further information on research design is available in the Nature Research Reporting Summary linked to this article.

## Data availability

The scRNAseq data and scTCRseq data generated in this study have been deposited in the GEO database under accession code GSE173303. The human reference genome data (GRCh38, v3.0.0, from 10X Genomics) are available at https://www.ncbi.nlm.nih.gov/assembly/GCF_000001405.26/. The remaining data are available within the article, Supplementary Information or Source Data file. Source data are provided with this paper.

## Code availability

All codes have been deposited to Github [https://github.com/Stnhy1/AR].

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

## Acknowledgements

This work was supported by The University of Texas MD Anderson Cancer Center Division of Internal Medicine Developmental Funds (S.T.K.), National Institutes of Health (NIH) R01 grants (R.N.: R01HL141966 and R01HL143520), NIH K08 grant (S.T.K.: K08AR079587), NIH K01 grant (N.A.-W.: K01AI163412), Cancer Prevention and Research Institute of Texas grant (R.N.: RP190326), Wilkes Melanoma Foundation grant (S.T.K., R.N., P.W., and A.D.), NIH/NCI through The University of Texas Lung Specialized Program of Research Excellence (SPORE) grant (T.C. and D.L.G.: 5P50CA070907), NIH/NCI Melanoma SPORE grant (R.N. and M.A.D.: P50CA221703), Dr. Miriam and Sheldon G. Adelson Medical Research Foundation (M.A.D.), the Anne and John Mendelsohn Chair for Cancer Research (M.A.D.), American Society of Clinical Oncology (ASCO) Career Development Award (T.C.: Project ID 12895), Khalifa Bin Zayed Al Nahyan Foundation (T.C.), the Welch Distinguished Chair Funds (A.F.), Institutional Research Grant (IRG) (S.T.K. and R.N.), the start-up research fund by the University of Texas MD Anderson Cancer Center (L.W.), and NIH core grants (Flow Cytometry Core Facility and ORION Core Facility: P30CA16672; Advanced Technology Genomics Core Facility: CA016672 and 1S10OD024977-01). J.K. is supported by the CPRIT Reseacrh Training Award CPRIT Training Program (RP210028). We are grateful to the Research Medical Library at The University of Texas MD Anderson Cancer Center for editing this manuscript.

## Author contributions

S.T.K. performed experiments, analyzed data, and wrote the manuscript; Y.C., M.M., M.B., N.A.-W., and Z.H. analyzed data; S.T.K., M.S.-A., J.H.T., H.L., J.W., A.Y.S., N.M.T., M.T.C., D.L.G., T.C., C.L., G.R.B., M.A., B.L., V.V., M.E.L., J.T, S.N.W., A.N., H.A.T., P.H., I.C.G., M.A.D., S.P.P., and A.D. identified and managed the cases; S.T.K., M.S.-A., J.H.T., and H.L. diagnosed arthritis-irAE and provided samples; J.K. and E.R. performed experiments and discussed results; M.S.-A., S.S.N., G.G.-M., P.H., A.F., J.Z., and A.D. analyzed the data and discussed the results; L.W. and R.N. oversaw the study, analyzed the data, and discussed the results; all authors reviewed and edited the manuscript.

## Competing interests

M.S.-A. has served as a consultant for Gilead, Avenue Therapeutics, ChemoCentryx, Pfizer, Eli Lilly and Bristol Myers Squibb. N.A.-W. has received honoraria for serving on a scientific advisory board for ChemoCentryx and served as a consultant for Chemo-Centryx. S.S.N. has received personal fees from Kite, a Gilead Company, Merck, Bristol Myers Squibb, Novartis, Celgene, Pfizer, Allogene Therapeutics, Sellas Life Sciences, Cell Medica/Kuur/Athenex, Incyte, Precision Biosciences, Legend Biotech, Adicet Bio, Calibr, Unum Therapeutics, Bluebird Bio, and Sana Biotechnology; research support from Kite, a Gilead Company, Bristol Myers Squibb, Merck, Poseida, Cellectis, Celgene, Karus Therapeutics, Unum Therapeutics, Allogene Therapeutics, Precision Biosciences, Acerta, and Adicet Bio; royalties from Takeda Pharmaceuticals; and has intellectual property related to cell therapy. M.T.C. has received honoraria for serve on a Scientific Advisory Board for Astellas, Eisai, EMD Serono, Exelixis, Genentech, Pfizer, Seattle Genetics, served as a consultant for ApricityHealth, Exelixis, Pfizer, non-branded educational programs supported by Bristol Myers Squibb, Exelixis, Merck, Pfizer, Roche, and research funding for clinical trials from ApricityHealth, AstraZeneca, EMD Serono/Pfizer, Exelixis, and Janssen. D.L.G. declares advisory board work for Janssen, Astra-Zeneca, GlaxoSmithKline and Sanofi. D.L.G. receives research grant funding from AstraZeneca, Janssen, Astellas, Ribon Therapeutics, NGM Therapeutics and Takeda. M.A. declares advisory board work for GlaxoSmithKline, Shattuck Lab, Bristol Myers Squibb, AstraZeneca. M.A. has received speaker fees from AstraZeneca, and Nektar Therapeutics. M.A. received research funding from Genentech, Nektar Therapeutics, Merck, GlaxoSmithKline, Novartis, Jounce Therapeutics, Bristol Myers Squibb, Eli Lilly, Adaptimmune, and Shattuck Lab. S.N.W. received research grants to the institution from AstraZeneca, Clovis Oncology, GSK/Tesaro, Roche/Genentech, Novartis, Cotinga Pharmaceuticals, Bayer, Bio-Path, and ArQule, and received consulting fees from AstraZeneca, Clovis Oncology, GSK/Tesaro, Roche/Genentech, Novartis, Merck, Pfizer, Eisai, Zentalis, Circulogene, and Agenus. A.N. received research grants to the institution from NCI, EMD Serono, MedImmune, Healios Onc. Nutrition, Atterocor/Millendo, Amplimmune, ARMO BioSciences, Karyopharm Therapeutics, Incyte, Novartis, Regeneron, Merck, Bristol- Myers Squibb, Pfizer, CytomX Therapeutics, Neon Therapeutics, Calithera Biosciences, TopAlliance Biosciences, Eli Lilly, Kymab, PsiOxus, Arcus Biosciences, NeoImmuneTech, ImmuneOncia, Surface Oncology, Monopteros Therapeutics, BioNTech SE, Seven & Eight Biopharma, and SOTIO Biotech AG. M.A.D. has been a consultant to Roche/Genentech, Array, Pfizer, Novartis, BMS, GSK, Sanofi-Aventis, Vaccinex, Apexigen, Eisai, and ABM Therapeutics, and he has been the PI of research grants to MD Anderson by Roche/Genentech, GSK, Sanofi-Aventis, Merck,

Myriad, and Oncothyreon. T.C. has received speakers' fees from the Society for Immunotherapy of Cancer, Bristol Myers Squibb, Roche, Medscape Oncology and PeerView Institute for Medical Education; reports consulting/advisory role fees from MedImmune, AstraZeneca, Bristol Myers Squibb, Merck & Co., Genentech, Arrowhead Pharmaceuticals and EMD Serono; reports institutional clinical research funding from Boehringer Ingelheim, MedImmune, AstraZeneca, EMD Serono, and Bristol Myers Squibb. The remaining authors declare no competing interests.

## Additional information

[1]Section of Rheumatology and Clinical Immunology, Department of General Internal Medicine, The University of Texas MD Anderson Cancer Center, Houston, TX 77030, USA. [2]Department of Genomic Medicine, The University of Texas MD Anderson Cancer Center, Houston, TX 77030, USA. [3]Department of General Internal Medicine, Baylor College of Medicine, Houston, TX 77030, USA. [4]Department of Immunology, The University of Texas MD Anderson Cancer Center, Houston, TX 77030, USA. [5]Department of Biology, Georgetown University, Washington, DC 20057, USA. [6]Department of Infectious Disease, Infection Control and Employee Health, The University of Texas MD Anderson Cancer Center, Houston, TX 77030, USA. [7]Department of Lymphoma and Myeloma, The University of Texas MD Anderson Cancer Center, Houston, TX 77030, USA. [8]Department of Genitourinary Medical Oncology, The University of Texas MD Anderson Cancer Center, Houston, TX 77030, USA. [9]Department of Thoracic Head and Neck Medical Oncology, The University of Texas MD Anderson Cancer Center, Houston, TX 77030, USA. [10]Department of Breast Medical Oncology, The University of Texas MD Anderson Cancer Center, Houston, TX 77030, USA. [11]Department of Neuro-Oncology, The University of Texas MD Anderson Cancer Center, Houston, TX 77030, USA. [12]Department of General Oncology, The University of Texas MD Anderson Cancer Center, Houston, TX 77030, USA. [13]Department of Gynecologic Oncology and Reproductive Medicine, The University of Texas MD Anderson Cancer Center, Houston, TX 77030, USA. [14]Department of Investigational Cancer Therapeutics, The University of Texas MD Anderson Cancer Center, Houston, TX 77030, USA. [15]Department of Leukemia, The University of Texas MD Anderson Cancer Center, Houston, TX 77030, USA. [16]Department of Melanoma Medical Oncology, The University of Texas MD Anderson Cancer Center, Houston, TX 77030, USA. [17]Department of Rheumatology and Rehabilitation, Assiut University Hospitals, Faculty of Medicine, Assiut University, El Fateh, Egypt. [18]Department of Laboratory Medicine, The University of Texas MD Anderson Cancer Center, Houston, TX 77030, USA. [19]The University of Texas MD Anderson Cancer Center UTHealth Graduate School of Biomedical Sciences (GSBS), Houston, TX 77030, USA. [20]Present address: H. Lee Moffitt Cancer Center and Research Institute, Tampa, FL 33612, USA. [21]These authors contributed equally: Sang T. Kim, Yanshuo Chu. ✉email: lwang22@mdanderson.org; rnurieva@mdandesron.org

