## [Peer Review File · Nature Communications]

Reviewers' Comments:

Reviewer #1:

Remarks to the Author:

Kim and colleagues have undertaken an in-depth analysis of immunologic features of a relatively rare irAE i.e. inflammatory arthritis by comprehensively analyzing PB and or synovial fluid samples from 20 patients and a comparator population of patients with non-inflammatory arthritis.

The strengths of this study are the size of the cohort and the detailed nature of the analytics including a number of functional assays in addition to the RNA-seq, detailed flow and multiplex technologies employed

At the same time the limitations are a small sample which upon partitioning for time of onset, concomitant therapies (glucocorticoids, DMARDs, Biologics), background heterogeneity of CPI regimens all lead to diminutive subsets to draw upon.

Overall however these data when compared to the extant literature on spontaneous and non-arthritic irAE are important and of interest.

The following should be considered to strengthen the presentation

1. I had a hard time following the progressive subsets of patients in terms of timing, concomitant therapies, and anti inflammatory therapies and beyond. While all of this is in the supplementary data I suggest some sort of flow diagram which outlines the cohorts and when they were sampled.

2. I have little to say about the presentation of the data which is strong --otherwise than to argue that p 11 the heading CXCL16 secreted by myeloid cells, play an important part is problematic. I believe this is not only discussion not data but ignores the possibility that CXCL16 on non hematopoietic cells i.e. synovial resident cells may also be playing an a prominent role as evidenced by work in both RA and its preclinical models. (this can be noted in discussion)

The remaining aspects of the paper are in the interpretation of these data which in general is fair and balanced and relevant citing high impact comparator studies. Unexplained in the discussion is the curious increased Treg functional signal in the setting of uncontrolled inflammation. Lack of relevance for the functional assay perhaps

Thoughts on discussion

P17 conjecture about the role of gamma IFN and B cell isotype switching is curious in this largely seronegative population.

The comparisons to colitis irAE data is strong

The seeming resistance of IL17 to glucocorticoids is an important and well discussed point

Reviewer #2:

Remarks to the Author:

Understanding the mechanistic basis for immune related adverse events such as arthritis-irAE is an important outstanding question. In this manuscript, Kim et al studied an impressively large cohort of arthritis-irAE patients given immune checkpoint blockade immunotherapy. They analyzed both blood (PB) and synovial fluid (SF) from these patients using flow cytometry and single-cell sequencing to describe cellular compositions and profiles of blood and SF from these patients. While some comparative analysis is performed most of the study focuses on describing overall profiles of PB and SF cells from arthritis-irAE. For each of these descriptions the authors typically conclude possible importance for irAE, however, nearly no definite linkages are provided. In addition, I object to the conclusiveness of a number of statements made throughout the paper. While these data are informative and might be resourceful given the uniqueness of these samples, I am concerned about the lack of clear conclusions made by this study.

Specific comments:

Minor: Fig. 1. How were neutrophils assessed in PBMCs. Are the authors concerned about inefficient encapsulation with 10x for the SF samples?

While it is informative IFN-g is the most commonly expressed cytokine among SF T cells, this does not really suggest a critical role of Th1/Tc1 cells in arthritis-irAE. By this logic, anything descriptive of SF could be critical in arthritis-irAE. That IFN-g is the most abundant cytokine in PMA+iono stimulated PBMC T cells is a trivial result and would likely be the same in any healthy donor. For Fig. 2g, stimulation with PMA+Iono should be mentioned in the results section.

Minor: It would be helpful if Fig. 2e and 2f were vertically aligned by cluster.

I am concerned about the choice of words in the section about T cell "dynamics". Dynamics can't really be investigated with a single time-point.

That there are expanded clones within Treg, MAIT, and gd T cells does not prove a critical role for these cells in arthritis-irAE (sentence starting on line 267 should be updated).

Clonal sharing between PB and SF does not prove progenitor-progeny relationships between phenotypic clusters. To argue that effector memory and terminally differentiated effector memory T cells do not return to the blood, the author reference a review from 2004. This statement should be qualified and primary reference(s) provided. It is probably not surprising that the most expanded T cell clonotypes from the blood are also more easily detected in the SF. Many of these could represent non-specific bystander T cell infiltration into the inflamed SF.

That SF myeloid cells express CXCL9/10/11 and T cells express CXCR3 and CXCR6 does not justify this sentence in the abstract: "Blood CX3CR1hi CD8+ T cell migration into joints is mediated by CXCL9/10/11/16 expressed by myeloid cells." While this is valid hypothesis, the data do not necessarily show this is the case.

Reviewer #3:

Remarks to the Author:

To the authors: The manuscript is a comprehensive characterization of the immune signature of immune-related arthritis induced by checkpoint inhibitor therapy against cancer. The authors have investigated and elucidated some key immune signatures and interactions in immune-related arthritis, a disease unique to a severe side effect of cancer immunotherapy. As the field of cancer immunotherapies is evolving to more effectively treat and cure cancers, we also must consider the adverse effects of these immunotherapies and determine how to prevent these negative manipulations to the immune system resulting in further disease. I have a few points of clarification and comments to improve the overall presentation and understandability of this study:

1. Referring to line 131: Since recent research has shown that arthritis is a systemic and immune related disease, another negative control for the PB samples should be included: PB from ICI-naïve patients with osteoarthritis to show that it is indeed ICI that is creating this distinct immunological signature separate from developing osteoarthritis itself.
2. Please provide a legend (or explanation in the figure caption) for the color coding of figures 1a, 1d, 2a, 2c, 2f, 3a, 4a (UMAP).
3. The description for the analysis of figure 4b is convoluted. What defines a T cell clone in this paper if it is not defined by TCR sequence? The description of these data in the figure caption are confusion and impossible to follow: "Table plots to (underlined word should be deleted) show shared top 100 expanded T cell clones across all subclusters. Data were totaled from all patients. Numbers indicate the shared top 100 T cell clones of each cluster pair." How can the numbers in the table plot indicate the shared top 100 T cell clones of each cluster pair and have numbers greater than 100? Are the PB and SF samples pooled together to compare the TCR clonotypes in each subcluster?
4. The main data referred to in the "Th17/Tc17 cells are steroid resistant" (line 362) section (extended data figure 7) should be brought into the main figures (potentially figure 5?) since it is a key finding of this paper.

Minor points:

1. Line 347 – specify that the amount of soluble cytokines is elevated in the SF supernatant. As the sentence reads it appears that the SF supernatant is elevated.
2. Line 365 – do you mean steroids hinder antitumor immunity revived by the ICIs? Instead of

harness?

Reviewer #4:

Remarks to the Author:

This manuscript describes in Depth subsets of leukocytes in peripheral blood and synovial fluid of patients experiencing arthritis as an immune-mediated secondary side effect of checkpoint inhibitors. State-of-the-art scRNAseq, flow cytometry and cytokine concentration determination are performed. Technically meritorious is the assessment of suppressor activity in very difficult synovial fluid samples. Authors state towards the end of the lengthy and speculative discussion the limitations of the study: sample size and heterogeneity, lack of data on the tumors requiring checkpoint immunotherapy, prior exposure to steroids in a good number of patients.

Comments and questions:

1. Are the monoclonal antibodies used in treatment reaching the synovial fluid? Can they be detected. Are they bound to T cells residing there? Which T cells subsets are expressing PD-1 and or CTLA-4 at the mRNA and protein level in peripheral blood and synovial fluid? Level of receptor occupancy?
2. Were cell samples frozen and thawed. What is the effect on this of this in terms of recovery of myeloid and lymphocyte subsets? Particularly with regard to synovial fluid samples?
3. What is the relevance of comparing peripheral blood and synovial fluid? What conclusions can be drawn?
4. Any differences regarding from patients treated with PD-1 versus Ipi + nivo? Should not Ipilimumab treatment lead to decreases in Tregs either in number or function?
5. Was arthritis or its treatment associated with cancer evolution? there is mounting evidence that IL-6 and TNF are actually protumor cytokines for the most part, so the neutralization of these cytokines could be actually helpful.
6. Any association with HLA-B27? Any resemblance to reactive arthritis?
7. Any other arthritis mediated by IFNg in the context of Th17? Comparisons with published literature would be useful to provide context.
8. What is known about paraneoplastic arthritis pathogenesis? Any resemblance? Better clinical description of the arthritis cases would be very useful (polyarthritis, oligoarthritis, symmetric arthritis, etc).
9. Supplementary table 1 does not report on cutaneous side effects, either because they were not present (that would be very odd) or because they were ignored. If existing do rashes correlate with arthritis?
10. There is much speculation and emphasis regarding mechanisms when merely based on not that strong correlative data. This aspect is more conspicuous regarding chemokines, chemokine receptors and chemoattraction.
11. Would treatment with checkpoint inhibitors exacerbate collagen induced arthritis in mice? Do the model recapitulate the human findings at the molecular and cellular level? This would probably permit incisive experimentation on the matter.

RESPONSES TO REVIEWERS' COMMENTS

We thank all reviewers for their critical comments and helpful advice. We have modified the manuscript according to the recommendations. Changes are indicated by track changes in the revised manuscript. Below are the answers addressing the critiques raised by the reviewers:

Reviewer #1 (Remarks to the Author): Expert in arthritis / irAE

Kim and colleagues have undertaken an in-depth analysis of immunologic features of a relatively rare irAE i.e. inflammatory arthritis by comprehensively analyzing PB and or synovial fluid samples from 20 patients and a comparator population of patients with non-inflammatory arthritis.

The strengths of this study are the size of the cohort and the detailed nature of the analytics including a number of functional assays in addition to the RNA-seq, detailed flow and multiplex technologies employed

At the same time the limitations are a small sample which upon partitioning for time of onset, concomitant therapies (glucocorticoids, DMARDs, Biologics), background heterogeneity of CPI regimens all lead to diminutive subsets to draw upon.

Overall however these data when compared to the extant literature on spontaneous and non-arthritic irAE are important and of interest.

Answer

We appreciate reviewer 1's comments emphasizing the strengths and importance of our study. As pointed out, to our best knowledge, the size of the cohort in this study (n=20) is the largest in arthritis-irAE research; while the sample size is small to control for confounders, our results guide us for further research in this field. In the future studies, collaborating with other institutes, we plan to collect and analyze more numbers of specimen from patients with arthritis-irAE.

The following should be considered to strengthen the presentation

1. I had a hard time following the progressive subsets of patients in terms of timing, concomitant therapies, and anti-inflammatory therapies and beyond. While all of this is in the supplementary data I suggest some sort of flow diagram which outlines the cohorts and when they were sampled.

Answer

We agree with the reviewer's suggestion. We added the new diagram flow demonstrating arthritis-irAE disease outcome and sample collections (Extended Data Figure 1a in the revised manuscript). Accordingly, we also revised the text (Page 6, Line 128-133 in the marked copy)

2. I have little to say about the presentation of the data which is strong --otherwise than to argue that p 11 the heading CXCL16 secreted by myeloid cells, play an important part is problematic. I believe this is not only discussion not data but ignores the possibility that CXL16 on non-hematopoietic cells i.e. synovial resident cells may also be playing a prominent role as evidenced by work in both RA and its preclinical models. (this can be noted in discussion)

Answer

We thank the constructive comment and agree with the reviewer that non-hematopoietic cells would also be a source of CXCL16 (Ruth et al. Arthritis and Rheumatism. 2006. PMID: 16508941; Nanki et al. Arthritis and Rheumatism. 2005. PMID: 16200580).

In response to the reviewer's point, we have revised the discussion (Page 20, Line 508-510 in the marked copy).

The remaining aspects of the paper are in the interpretation of these data which in general is fair and balanced and relevant citing high impact comparator studies. Unexplained in the discussion is the curious increased Treg functional signal in the setting of uncontrolled inflammation. Lack of relevance for the functional assay perhaps

Answer

[Redacted]

Interestingly, studies analyzing SF and pairing PB samples from rheumatoid arthritis (RA) or Juvenile Idiopathic Arthritis (JIA) patients revealed that SF Tregs were more suppressive compared with matching PB Tregs (Amersfoort et al. Arthritis and Rheumatism. 2004. PMID: 15457445; de Kleer et al. Journal of Immunology. 2004. PMID: 15128835). It would be an important topic in the future to understand the biology of Tregs in arthritis-irAE versus classical autoimmune arthritis. In addition, recent study has showed that Treg functions might be inversely correlated with severity of irAEs, suggesting heterogeneity of Treg functions in irAEs (Kim et al. Oncoimmunology. 2020. PMID: 32076579). In this regard, it would also be a critical future topic to investigate Treg biology between arthritis-irAE and non-arthritic irAEs.

In response to the reviewer's comment, we revised the manuscript to discuss enhanced SF Treg functions (Page 19, Line 467-484; Page 20, Line 485-487 in the marked copy).

[Redacted]

Thoughts on discussion
P17 conjecture about the role of gamma IFN and B cell isotype switching is curious in this largely seronegative population.

[Redacted]

The comparisons to colitis irAE data is strong

Answer

We appreciate the reviewer's comment. We agree that understanding of altered immunity of irAEs based on involved organs (joint vs. colon, for example) would be another important topic in irAE researches.

The seeming resistance of IL17 to glucocorticoids is an important and well discussed point

Answer

Thank you very much for the comment. Indeed, the reviewer 3 had the same impression. In response to the comment of the reviewers 1 and 3, we moved the extended data figure 7 to the main figure 6 in the revised manuscript. We also revised the text accordingly (Page 17-19 [figure part labels]; Page 47, Line 1189-1200 in the marked copy).

Reviewer #2 (Remarks to the Author): Expert in scRNA immunology

Understanding the mechanistic basis for immune related adverse events such as arthritis-irAE is an important outstanding question. In this manuscript, Kim et al studied an impressively large cohort of arthritis-irAE patients given immune checkpoint blockade immunotherapy. They analyzed both blood (PB) and synovial fluid (SF) from these patients using flow cytometry and single-cell sequencing to describe cellular compositions and profiles of blood and SF from these patient. While some comparative analysis is performed most of the study focuses on describing overall profiles of PB and SF cell from arthritis-irAE. For each of the these descriptions the authors typically conclude possible importance for irAE, however, nearly no definite linkages are provided. In addition, I object to the conclusiveness of a number of statements made throughout the paper. While these data are informative and might be resourceful given the uniqueness of these samples, I am concerned about the lack of clear conclusions made by this study.

Answer

We appreciate the reviewer's valuing our findings and constructive comments. We acknowledge that there are limitations to draw conclusions in this study, mainly due to the nature of translational research. During the revision period, we performed extensive literature searches, re-analyses original data, and new experiments (Comments' Response Figures 1-11) to provide additional linkages to the conclusions we proposed. We would also like to emphasize that is the first study unveiling immune landscape of patients with arthritis-irAE utilizing patients' samples. Since we do not have pre-clinical mouse models of arthritis-irAE, it is critical to get the most of data from the patients with arthritis-irAE. Such data will not only help us to understand arthritis-irAE deeper, but also to provide scientific blue prints to generate the arthritis-irAE models. Indeed, building from the insights of this manuscript, we are generating murine arthritis-irAE model and expect to publish in near the future. We think that this model will tremendously help us to elucidate mechanisms of the arthritis-irAE.

Specific comments:

Minor: Fig. 1. How were neutrophils assessed in PBMCs. Are the authors concerned about inefficient encapsulation with 10x for the SF samples?

Answer

Thank you for the insightful comments. Neutrophils in PBMCs were accessed by Ficoll gradient techniques, during which neutrophils in PB might have been lost. Having said that though, based on the data for gene expression, the fraction reads in cells scores were between 85.3% and 96.8%, suggesting that the partitioning worked well for the samples. Further, we detected the cell clusters with neutrophil gene profiles (Supplementary Table 3) and confirmed the presence of neutrophils in SF and PB with flow cytometry (Extended Data Figure 1). Finally, we would like to emphasize that T cells, the main focus in our manuscript, have been stable during cell processes (Comments' Response Figure 9).

While it is informative IFN-g is the most commonly expresses cytokine among SF T cells, this does not really suggest a critical role of Th1/Tc1 cells in arthritis-irAE. By this logic, anything descriptive of SF could be critical in arthritis-irAE. That IFN-g is the most abundant cytokine in PMA+iono stimulated PBMC T cells is a trivial result and would likely be the same in any healthy donor. For Fig. 2g, stimulation with PMA+Iono should be mentioned in the results section.

Answer

We appreciate the reviewer's comment. We re-visited the clinical data of our cohort with longitudinal samples (before and after arthritis therapy). Clinical disease activity index, the clinical parameter widely used to measure arthritis disease activity (Anderson et al. Arthritis Care and Research. PMID: 22473918), were decreased after the arthritis therapy (Comments' Response Figure 3) as is Th1/Tc1 cells (Figure 6c-d in the revised manuscript). Therefore, it is speculated that Th1/Tc1 cells might play a pathogenic role in arthritis-irAE.

Comments' Response Figure 3. Clinical Disease Activity Index (CDAI) of the arthritis-irAE patients (n=7) whose blood samples were collected and analyzed before and after arthritis treatment. Paired t-test. **P<0.01.

We would also like to emphasize that IFN γ has been getting attentions for their role in the pathogenesis of autoimmune arthritis including rheumatoid arthritis. For example, recent study showed the enrichment IFN γ + CD8+ T cells in RA synovium (Zhang et al. Nature Immunology. 2019. PMID: 31061532). Same study also revealed that RA synovial fibroblasts highly express IFN γ -inducible protein 30. In addition, the meta-analysis across datasets of gene expression microarray demonstrated the pathogenic role of IFN γ in RA (Lee et al. Cytokine. 2020. PMID: 31881419).

Having said that though, we absolutely agree with the reviewer's concern that abundance does not necessarily guarantee scientific significance. Given the nature of translational researches in this study, there are limitations in investigating the biologic causality, and we are generating the arthritis-irAE murine model by modifying the collagen-induced arthritis mouse model. Utilizing the model, we will genetically and pharmacologically deplete IFN γ to address the role of IFN γ on arthritis-irAE.

In response to the reviewer's comment, we expanded the discussion to address the role of IFN γ on inflammatory arthritis (Page 18, Line 455-458 in the marked copy) and corrected description for Figure 2g (Page 9, Line 225-226; Page 44, Line 1123).

Minor: It would be helpful if Fig. 2e and 2f were vertically aligned by cluster.

Answer

We vertically aligned the figures as the reviewer suggested (Figure 2d-e in the revised manuscript).

I am concerned about the choice of words in the section about T cell "dynamics". Dynamics can't really be investigated with a single time-point.

Answer

Thank you for the clarification. “Dynamic” in our manuscript mainly means trajectory relationships between SF and PB T cells, but we certainly agree with the reviewer’s concern that “dynamic” generally contains the concept of multiple time-points. In response to the reviewer’s comment, we revised the sentences which contain “dynamic”. (Page 8, Line 180; Page 11, Line 270-272; Page 12, Line 299-301; Page 13, Line 320-322; Page 18, Line 448; Page 20, Line 488; Page 49, Line 1232 in the marked copy).

That there are expanded clones within Treg, MAIT, and gd T cells does not prove a critical role for these cells in arthritis-irAE (sentence starting on line 267 should be updated).

Answer

Thank you very much for the careful comment. We revised the sentences from the lines 265 to 268 in the original manuscript (Page 11, Line 277-278; Page 12, Line 279 in the marked copy).

Clonal sharing between PB and SF does not prove progenitor-progeny relationships between phenotypic clusters. To argue that effector memory and terminally differentiated effector memory T cells do not return to the blood, the author reference a review from 2004. This statement should be qualified and primary reference(s) provided. It is probably not surprising that the most expanded T cell clonotypes from the blood are also more easily detected in the SF. Many of these could represent non-specific bystander T cell infiltration into the inflamed SF.

[Redacted]

We certainly agree with the reviewer that the most expanded T cell clones could be bystander T cells. However, at the same time, it is also possible that these clones might be self-reactive and/or tumor-specific T cells. Indeed, the TCR analyses of melanoma patients who died due to myositis-myocarditis after ICI therapy revealed the presence of T cell clones co-existing in inflamed myocardium, skeletal muscle, and melanoma tumor, raising the possibility that antigens (epitopes) exist in the myocardium, skeletal muscle, and perhaps tumors which were recognized by the same T-cell clones (Johnson et al. NEJM. 2017. PMID: PMID: 27806233). Again, we

absolutely agree with the reviewer that whether irAEs are antigen-specific responses or not would be one of the foremost important topic in irAE research. Indeed, we are addressing antigen-specificity in arthritis-irAE by characterizing self-reactive and/or tumor specific T cells from arthritis-irAE patients with positive HLA-DRB1 04:01 utilizing tetramers loaded with known autoantigens and tumor-antigens (Snir et al. Arthritis and Rheumatology. 2011. PMID: 21567378) (Zarour et al. Cancer Research. 2000. PMID: 10987311).

We deeply appreciated for the reviewer's clarifying comment "To argue that effector memory and terminally differentiated effector memory T cells do not return to the blood, the author reference a review from 2004. This statement should be qualified and primary reference(s) provided". Multiple studies demonstrated that inflamed tissues including lung, brain, and skin are enriched with Tem cells, mostly CD8+ effector memory (CD8+ Tem) and terminally differentiated effector memory (CD8+ Temra), suggesting that these tissue-infiltrated cells may not be returned to the blood (Gebhardt et al. Nature. 2011. PMID: 21841802; Ely et al. Journal of Immunology. 2006. PMID: 16365448; Wakim. PNAS. 2010. PMID: 20923878). We provided these references in revised version (Reference numbers: 30,31, and 32 in the marked copy). Also, since it is controversial whether this can be applied to CD4+ T cells as well as final destiny of CD8+ Tem and CD8+ Temra cells have not been entirely proven, we revised the sentence highlighted by the reviewer (Page 12, Line 292-299 in the marked copy).

[Redacted]

That SF myeloid cells express CXCL9/10/11 and T cells express CXCR3 and CXCR6 does not justify this sentence in the abstract: “Blood CX3CR1hi CD8+ T cell migration into joints is mediated by CXCL9/10/11/16 expressed by myeloid cells.” While this is valid hypothesis, the data do not necessarily show this is the case.

Answer

We appreciate the reviewer’s critical comment. In response to the reviewer’s comment, we performed a migration assay with the method we published in the past (Kim et al Journal of Immunology. 2018. PMID: 30030323) (Comments’ Response Figure 5a). Briefly, we sorted naïve CD8+ T cells (CD3+ CD8+ CD45RA+ CX3CR1lo) and CX3CR1hi effector CD8+ T cells (CD3+ CD8+ CD45RA- CX3CR1hi) from PB samples of patients with arthritis-irAE (n=6) using FACs Aria (Comments’ Response Figure 5b). Sorted cells were washed and resuspended in migration medium (RPMI 1640 with 0.5% fetal bovine albumin) at 50×10^3 cells/ml. A total of 25×10^3 cells were loaded in a 8-mm pore transwell (BD Falcon, Cat#: Costar 3464). In the lower chamber, 500 ul of migration medium was placed in the presence/absence of 1 ug/mL of CXCL9 (R&D Systems, Cat #: 392-MG), CXCL10 (R&D Systems, Cat #: 266-IP), CXCL11 (R&D Systems, Cat #: 672-IT), and CXCL16 (R&D Systems, Cat #: 976-CX). Cells were allowed to migrate at 37°C for 3 hours. Subsequently, the cells in the lower chamber were collected and loaded for flow cytometry. Just before samples were loaded, 20×10^3 of beads (Spherotech) were added into each tube to standardize cell numbers between tubes. Events were analyzed with FlowJo software to calculate the ratio of migrated cells in response to the chemokines to migrated cells in absence of chemokines (Comments’ Response Figure 5c-d). As expected, CD45RA+ naïve CD8+ T cells did not migrated well in response to CXCL9/10/11/16 (ratio relative to absence of CXCL9/10/11/16, mean±SD, 1.04 ± 0.14 , P=0.48) In contrast, CX3CR1hi effector CD8+ T cells migrated well in response to CXCL9/10/11/16 (ratio relative to absence of CXCL9/10/11/16, 1.90 ± 0.34 , P=0.001).

Comments' Response Figure 5. CX3CR1hi effector CD8+ T cells migrated in response to CXCL9/10/11/16. (a) Schematics showing the experimental designs **(b)** Flow cytometry plots showing sorting strategies. **(c)** After allowing migration of naïve or CX3CR1hi effector CD8+ T cells for three hours in the presence/absence of CXCL9/10/11/16, the cells in the lower chamber were collected. Representative flow cytometry plots taken at the time when 5,000 magnetic beads were collected. Tn, naïve CD8+ T cells. **(d)** Ratio of migrated cells in response to CXCL9/10/11/16 to migrated cells in absence of the chemokines. n=6 PB samples from arthritis-irAE patients. Tn, naïve CD8+ T cells; CX3CR1hi, CX3CR1hi effector CD8+ T cells. Paired t-test. **P<0.01

In addition to the migration assay, we also measured the levels of CXCL9/10/11/16 in serum and SF supernatant with commercially available ELISA (Cat #: EHCXCL9 and EHCXCL16, all

Invitrogen) and Multiplex (MSD U-Plex) kits (Comments' Response Figure 6). Levels of CXCL9, 10, and 16 were increased in SF supernatant compared with matching serum in arthritis-irAE group (Comments' Response Figure 6a). Levels of the SF CXCL9, 10, and 11 were higher in arthritis-irAE patients compared with OA patients (Comments' Response Figure 6b). Studies suggest that increase of serum CXCL9 after ICI therapy could be a biomarker to predict development of irAE, including arthritis-irAE (Gerber et al. British Journal of Cancer. 2018. PMID: 30377338). Consistent with the data, compared to patients without irAEs and ICI-naïve OA patients, serum level of CXCL9 were increased in arthritis-irAE group (Comments' Response Figure 6c).

Our data from scRNAseq (Figure 4c and Extended Data Figure 4 in revised manuscript), the migration assays (Comments' Response Figure 5), and ELISA/Multiplex (Comments' Response Figure 6) collectively suggests that Blood CX3CR1hi CD8+ T cell migration into joints is mediated by CXCL9/10/11/16 expressed mainly by myeloid cells. We deeply appreciate the reviewer's critical comment, and we put the data of the migration assays in the revised manuscript (Figure 4d and Extended Data Figure 5 in the revised manuscript). Accordingly, we also revised the text (Page 13, Line 309-322; Page 33, Line 843-845; Page 34, Line 846-856, Page 46, Line 1172-1174, Page 49, Line 1245-1250 in the marked copy).

Comments' Response Figure 6. Quantification of CXCL9/10/11/16 in serum and synovial fluid (SF) supernatant. (a) Comparison of the chemokines between serum and matching SF supernatant within individuals. OA, osteoarthritis; irAE, arthritis-irAE. Paired t-test. * $P<0.05$, **** $P<0.001$. (b) Comparison of the chemokines in SF supernatant between osteoarthritis (OA) and arthritis-irAE (irAE) patients. Unpaired t-test. * $P<0.05$, **** $P<0.001$. (c) Comparison of chemokines in serum from patients with osteoarthritis (OA), patients who well tolerated immune checkpoint inhibitors at least 12 weeks (no-irAE), and patients with arthritis-irAE (irAE). One-way ANOVA test. ** $P<0.01$.

Reviewer #3 (Remarks to the Author): Expert in osteoarthritis immunology

To the authors: The manuscript is a comprehensive characterization of the immune signature of immune-related arthritis induced by checkpoint inhibitor therapy against cancer. The authors have investigated and elucidated some key immune signatures and interactions in immune-related arthritis, a disease unique to a severe side effect of cancer immunotherapy. As the field of cancer immunotherapies is evolving to more effectively treat and cure cancers, we also must consider the adverse effects of these immunotherapies and determine how to prevent these negative manipulations to the immune system resulting in further disease. I have a few points of clarification and comments to improve the overall presentation and understandability of this study:

Answer

We appreciate the reviewer's summarizing our study nicely and valuing our observations. We certainly agree with the reviewer that as the immune-based cancer therapy is being explosively applied to various cancers, we will encounter more cases of patients with autoimmune/autoimmune-complications including inflammatory arthritis. We also thank for the reviewer's constructive comments.

1. Referring to line 131: Since recent research has shown that arthritis is a systemic and immune related disease, another negative control for the PB samples should be included: PB from ICI-naïve patients with osteoarthritis to show that it is indeed ICI that is creating this distinct immunological signature separate from developing osteoarthritis itself.

Answer

As the reviewer points out, recent studies show that irAEs, including arthritis-irAE, are systemic disorders. In response to the reviewer's comment, we repeated multiplex of inflammatory cytokines of serum samples including ICI-naïve osteoarthritis group as another negative control (Comments' Response Figure 7). Alike the reviewer's prediction, the levels of inflammatory cytokines were lower in ICI-naïve OA group compared to patients receiving ICIs (no-irAEs, PD-1 inhibitor arthritis, and combined ICI arthritis groups). Consistent with the result in our previous manuscript, although not reached statistical significance, we observed that the levels of serum IL-6 and IL-17A from combine ICI arthritis patients were higher than those from other three groups.

We updated supplemental figures (Extended Data Figure 1b and 7b in the revised manuscript). We also revised the text accordingly (Page 16, Line 388-393 in the marked copy).

Comments' Response Figure 7. Quantification of inflammatory cytokines in serum. One-way analysis of variance. *P<0.05, **P<0.01. Bars indicate the mean and SEM. No-irAEs; patients who had no irAEs for at least 12 weeks after initiating of immune checkpoint inhibitor (ICI) therapy; OA, osteoarthritis; P, PD-1 inhibitor arthritis; C, Combined ICI arthritis. IFN γ , interferon gamma; IL, interleukin.

2. Please provide a legend (or explanation in the figure caption) for the color coding of figures 1a, 1d, 2a, 2c, 2f, 3a, 4a (UMAP).

Answer

We provided a legend for the color coding of figure 1a, 1d, 2a, 2c, and 2f. We did not put the legend for figure 3a as it is eminent that cluster 1 is PB Tregs while cluster 2 is SF Tregs as shown in figure 3b. Because we have already use color coding of figure 4a to show that top 100 TCR clones from individual patients, in order to prevent confusion, we did not use color coding for clusters but showed the legend as a plain table.

In response to the reviewer's comment, we revised the figures 1-2 and text (figure part labels) accordingly (Page 8-10 in the marked copy).

3. The description for the analysis of figure 4b is convoluted. What defines a T cell clone in this paper if it is not defined by TCR sequence? The description of these data in the figure caption are confusion and impossible to follow: "Table plots to (underlined word should be deleted) show shared top 100 expanded T cell clones across all subclusters. Data were totaled from all patients. Numbers indicate the shared top 100 T cell clones of each cluster pair." How can the numbers in the table plot indicate the shared top 100 T cell clones of each cluster pair and have numbers greater than 100? Are the PB and SF samples pooled together to compare the TCR clonotypes in each subcluster?

Answer

We appreciate the reviewer's comment. Along with scRNAseq, we also performed paired scTCRseq with the Chromium Single Cell V(D)J Enrichment Kit, Human T cell on the same cDNA libraries. The phenotypes and genotypes of T cells were integrated by the unique cell barcodes. T cell clonotypes were defined by Cell Ranger based on the similarity of the paired TCRAV/TCRBV segments and the CDR3 sequences. We apologize that the Methods section describing scTCRseq did not provide adequate information. In the revised manuscript, we have updated the Methods to include more details.

In addition, we apologize for the careless mistake in the legend of Main Figure 4b, and we have make corrections.

In response to the reviewer's comment, we revised/added some sentences (Page 27, Line 670-673; Page 30, Line 763-770; Page 31, Line 771-776; Page 46, Line 1158-1174 in the marked copy).

4. The main data referred to in the "Th17/Tc17 cells are steroid resistant" (line 362) section (extended data figure 7) should be brought into the main figures (potentially figure 5?) since it is a key finding of this paper.

Answer

We appreciate the reviewer for valuing the data from longitudinal PB sample analyses. Indeed, the reviewer 1 raised the same point. We moved the extended data figure 7 to main figure 6 (in revised manuscript).

Minor points:

1. Line 347 – specify that the amount of soluble cytokines is elevated in the SF supernatant. As the sentence reads it appears that the SF supernatant is elevated.

Answer

We agree with the reviewer's concern and revised the text (Page 15, Line 376-379 in the marked copy).

2. Line 365 – do you mean steroids hinder antitumor immunity revived by the ICIs? Instead of harness?

Answer

We thank for the reviewer's clarifying comment. We changed "harness" to "hinder" (Page 16, Line 401 in the marked copy).

Reviewer #4 (Remarks to the Author): expert in Cancer immunology/immunotherapy

This manuscript describes in Depth subsets of leukocytes in peripheral blood and synovial fluid of patients experiencing arthritis as an immune-mediated secondary side effect of checkpoint inhibitors. State-of-the-art scRNAseq, flow cytometry and cytokine concentration determination are performed. Technically meritorious is the assessment of suppressor activity in very difficult synovial fluid samples. Authors state towards the end of the lengthy and speculative discussion the limitations of the study: sample size and heterogeneity, lack of data on the tumors requiring checkpoint immunotherapy, prior exposure to steroids in a good number of patients.

Answer

We thank the reviewer for enlightening strengths/novelty of our study.

Comments and questions:

1. Are the monoclonal antibodies used in treatment reaching the synovial fluid? Can they be detected. Are they bound to T cells residing there? Which T cells subsets are expressing PD-1 and or CTLA-4 at the mRNA and protein level in peripheral blood and synovial fluid? Level of receptor occupancy?

Answer

We appreciate the reviewer's insightful comment. During the revision, we had multiple limitations that preclude us from performing this assay: 1) limited number of available methods to detect ICI-bound T cells; 2) none of protocol designed to detect ICI-bound T cells in synovial fluid; and 3) limited sample numbers to perform assay. We are working on designing the protocol that will be used in our further experiments. Again, we certainly agree with the reviewer that these are very

critical questions and discuss the importance to investigate ICI-bound T cells in SF in the future studies (Page 22, Line 559-562; Page 23, Line 563-564 in marked copy).

To investigate expression of PD-1 and CTLA-4 on T cells at the mRNA and protein level, we revisited the scRNAseq data and performed flow cytometry to validate the findings (Comments' Response Figure 8). Clusters 4,5,8,9, and 12, which are preferentially present in SF, highly expressed CTLA-4; PD-1 were highly expressed on clusters 5,8,9, and 12 (Comments' Response Figure 8a). As expected, naïve T cells (clusters 1,2, and 6), which are mainly present in PB (Figure 2c in the revised manuscript), barely express both PD-1 and CTLA-4 (Comments' Response Figure 8a). To validate scRNAseq findings, we stained SF and matching PB samples to investigate expression and occupancy of PD-1 and CTLA-4 on PB CD45RA+ naïve T cells (analogue to clusters 1,2, and 6; served as a negative control), SF CD25hi FoxP3+ Tregs (cluster 4), SF CXCL13+ CD3+ T cells (cluster 5), SF CXCR3+ CD45RA- CD3+ T cells (clusters 8-9), and SF Ki67+ CD3+ T cells (cluster 12) (Comments' Response Figure 8b). Consistent with scRNAseq findings, compared with mean fluorescence intensity (MFI) of CTLA-4 on SF CD25hi FoxP3+ Tregs, SF CXCL13+ CD3+ T cells, SF CXCR3+ CD45RA- CD3+ T cells, and SF Ki67+ CD3+ T cells was higher than MFI of CTLA-4 on PB CD45RA+ naïve T cells (Comments' Response Figure 8d). Likewise, PD-1 was more expressed on SF CXCL13+ CD3+ T cells, CXCR3+ CD45RA- CD3+ T cells, and Ki67+ CD3+ T cells (Comments' Response Figure 8d). In harmony with MFI data, frequency of CTLA-4+ populations were increased in SF CD25hi FoxP3+ Tregs, SF CXCL13+ CD3+ T cells, SF CXCR3+ CD45RA- CD3+ T cells, and SF Ki67+ CD3+ T cells while frequency of PD-1+ populations were increased in SF CXCL13+ CD3+ T cells, CXCR3+ CD45RA- CD3+ T cells, and Ki67+ CD3+ T cells (Comments' Response Figure 8e-f).

f

Matching clusters on scRNAseq	Cells on flow cytometry	% of CTLA-4+ on flow cytometry (mean ± SD)
1,2,6	CD45RA+ CD3+ T	1.25 ± 0.96
4	CD25hi FoxP3+ Tregs	99.30 ± 1.40
5	CXCL13+ T	78.96 ± 15.33
8,9	CXCR3+ CD45RA- T	97.06 ± 2.67
12	Ki67+ T	47.38 ± 34.88

Matching clusters on scRNAseq	Cells on flow cytometry	% of PD-1+ on flow cytometry (mean ± SD)
1,2,6	CD45RA+ CD3+ T	1.22 ± 1.07
4	CD25hi FoxP3+ Tregs	12.09 ± 3.16
5	CXCL13+ T	52.54 ± 17.46
8,9	CXCR3+ CD45RA- T	75.14 ± 17.92
12	Ki67+ T	17.50 ± 10.64

Comments' Response Figure 8. Expression of CTLA-4 and PD-1 on T cells at the mRNA and protein level. (a) Bubble plot showing expression of CTLA-4 and PD-1. Legend describing clusters was shown in the right panel. (b) Peripheral blood (PB) and matching synovial fluid (SF) samples were stained to investigate expression and occupancy of CTLA-4 and PD-1. Representative flow cytometry plots. (c) Histogram to show frequencies of CTLA-4 and PD-1 expressing cells in the indicated cell population. Mean florescent intensity (MFI) is shown at the upper right corner as a red text. Numbers in the parentheses represent matching cell cluster(s) on scRNAseq. (d) Ratio of MFI of indicated cell populations to the MFI of PB naïve T cells. Paired t-test. *P<0.05, **P<0.01, ***P<0.001. (e) Frequencies of CTLA-4 or PD-1 expressing cells within indicated cell population. Numbers in the parentheses represent matching cell cluster(s) on scRNAseq. One-way ANOVA test. ***P<0.001. (f) Absolute numbers of the frequencies expressing CTLA-4 or PD-1 in the indicated cell populations.

2. Were cell samples frozen and thawed. What is the effect on this of this in terms of recovery of myeloid and lymphocyte subsets? Particularly with regard to synovial fluid samples?

Answer

Thank you for the critical comment. During the revision, we were able to collect two SF and matching PB from patients with arthritis-irAE (Comments' Response Figure 9a-b). Briefly, after collecting samples, half volume of the samples were stained right away and fixed. The other half were cryopreserved. Seven days later, the cryopreserved samples were thawed, washed, stained, and fixed. Both fresh and frozen samples were run at the same time and we analyzed major immune cell populations. Regarding SF samples, some of cells died during freezing (% of within total CD45+ cells; fresh vs. frozen; 92.96 ± 5.32 vs. 45.49 ± 3.78 ; $P= 0.01$), most of which appeared to be neutrophils (% of within total CD45+ cells; fresh vs. frozen; 64.20 ± 14.30 vs. 11.07 ± 4.81 ; $P= 0.08$). PB samples were stable during freezing/thaw (Comments' Response Figure 9c-d).

Although some of SF cells died during freezing/thaw, viability of $45.49 \pm 3.78\%$ seems to be reasonable. Importantly, both SF and PB T cells, main focus in the manuscript, were stable during freezing/thaw (% of SF CD4+ T cells within total CD45+ cells; fresh vs. frozen; 10.63 ± 4.57 vs. 10.80 ± 4.25 ; $P= 0.58$) (% of SF CD8+ T cells within total CD45+ cells; fresh vs. frozen; 4.21 ± 3.44 vs. 5.36 ± 3.82 ; $P= 0.15$) (% of PB CD4+ T cells within total CD45+ cells; fresh vs. frozen; 22.04 ± 3.85 vs. 10.80 ± 4.25 ; $P= 0.54$) (% of PB CD8+ T cells within total CD45+ cells; fresh vs. frozen; 7.00 ± 5.68 vs. 6.98 ± 5.60 ; $P= 0.72$).

Given relatively acceptable recovery rates in both SF and PB samples as well as stability of T cells, the main points of the manuscript would not be altered even if cryopreserved samples were analyzed.

C

Fresh SF

Immune cell subsets	Markers
pDC	HLA-DR+ CD11c+
mDC	HLA-DR+ CD123+
Neutrophils	SSC-Ahi CD16+ CD24+
Macrophages (Mac)	CD14+
NK cells	CD3- CD56+
NK T cells	CD3+ CD56+
B cells	CD3- CD56- CD19+
$\gamma\delta$ T cells	CD3+ CD56- $\gamma\delta$ TCR+
CD4+ T cells	CD3+ CD56- $\gamma\delta$ TCR- CD4+
CD8+ T cells	CD3+ CD56- $\gamma\delta$ TCR- CD8+

Frozen SF

Immune cell subsets	Markers
pDC	HLA-DR+ CD11c+
mDC	HLA-DR+ CD123+
Neutrophils	SSC-Ahi CD16+ CD24+
Macrophages (Mac)	CD14+
NK cells	CD3- CD56+
NK T cells	CD3+ CD56+
B cells	CD3- CD56- CD19+
$\gamma\delta$ T cells	CD3+ CD56- $\gamma\delta$ TCR+
CD4+ T cells	CD3+ CD56- $\gamma\delta$ TCR- CD4+
CD8+ T cells	CD3+ CD56- $\gamma\delta$ TCR- CD8+

Fresh PB

Immune cell subsets	Markers
pDC	HLA-DR+ CD11c+
mDC	HLA-DR+ CD123+
Neutrophils	SSC-Ahi CD16+ CD24+
Macrophages (Mac)	CD14+
NK cells	CD3- CD56+
NK T cells	CD3+ CD56+
B cells	CD3- CD56- CD19+
$\gamma\delta$ T cells	CD3+ CD56- $\gamma\delta$ TCR+
CD4+ T cells	CD3+ CD56- $\gamma\delta$ TCR- CD4+
CD8+ T cells	CD3+ CD56- $\gamma\delta$ TCR- CD8+

Frozen PB

Immune cell subsets	Markers
pDC	HLA-DR+ CD11c+
mDC	HLA-DR+ CD123+
Neutrophils	SSC-Ahi CD16+ CD24+
Macrophages (Mac)	CD14+
NK cells	CD3- CD56+
NK T cells	CD3+ CD56+
B cells	CD3- CD56- CD19+
$\gamma\delta$ T cells	CD3+ CD56- $\gamma\delta$ TCR+
CD4+ T cells	CD3+ CD56- $\gamma\delta$ TCR- CD4+
CD8+ T cells	CD3+ CD56- $\gamma\delta$ TCR- CD8+

Comments' Response Figure 9. Comparison of major immune cell subsets from fresh or cryopreserved synovial fluid (SF) and peripheral blood (PB) samples. (a) Experimental design (b) Basic clinical information of arthritis-irAE patients (n=2) who provided SF and PB samples. (c) Representative flow cytometry plots of samples from SF (fresh and frozen) and PB (fresh and frozen) samples. (d) Proportion of immune cell subsets within CD45+ cells from SF (upper panel) and PB (lower panel). pDC, plasmacytoid dendritic cells; mDC, myeloid dendritic cells; NK, natural killer cells; $\gamma\delta$ T, gamma delta T cells. Paired t-test. *P<0.05.

3. What is the relevance of comparing peripheral blood and synovial fluid? What conclusions can be drawn?

Answer

As the reviewer 3 mentioned, the arthritis-irAE is systemic disorder. Therefore, we thought that by comparing SF and pairing PB from the patients with arthritis-irAE, we could get valuable insights of mechanisms underlying the arthritis-irAE. Similar to our approaches, by comparing SF and matching PB samples of psoriatic arthritis patients, one recent study provided important insights on pathogenesis of psoriatic arthritis (Penkava et al. Nat Communications. 2020. PMID: 32958743).

Focusing on our study, by comparing SF and pairing PB samples, we were able to conclude that

- Interferon gamma-producing T cells were enriched in synovial fluid from patients with arthritis-irAE.
- CX3CR1hi CD8+ T cells in blood and CXCR3hi CD8+ T cells in synovial fluid are clonally expanded, with significant overlap of their T cell repertoires.
- Interactions between T cells and myeloid cells mediated by CXCR3 and CXCR6 might play an important role in arthritis-irAE pathogenesis.
- Arthritis-irAE induced by combined immune checkpoint inhibitor therapy is steroid-resistant and exhibits enhanced Th17 cell signatures.

4. Any differences regarding from patients treated with PD-1 versus Ipi + nivo? Should not Ipilimumab treatment lead to decreases in Tregs either in number or function?

Answer

[Redacted]

We deeply appreciate the reviewer's suspicion that Treg functions might be altered in the patients with combined ICI arthritis. In response to the reviewer's comment, we revised the text in the revised manuscript (Page 20, Line 486-487 in the marked copy)

[Redacted]

5. Was arthritis or its treatment associated with cancer evolution? there is mounting evidence that IL-6 and TNF are actually protumor cytokines for the most part, so the neutralization of these cytokines could be actually helpful.

Answer

Thank you very much for the insightful comment. While there might be beneficial effects on survival with the use of IL-6 and/or TNF inhibitors, our sample size is too small to draw any meaningful or robust conclusions. We did analyze overall and progression-free survival of the patients with arthritis-irAE (n=20) over one-year follow-up period according to arthritis-irAE treatment received (Comments' Response Figure 11). Only one patient, who received TNFi died during follow up due to tumor progression. Patients receiving IL-6 inhibitor did not die or have progression but the numbers are too small for statistical comparisons.

Comments' Response Figure 11. Kaplan-Meier Analyses of Overall survival (a) and Progression-free survival (b). No IL-6i/TNFi, the group of arthritis-irAE patients who did not receive IL-6 inhibitor nor TNF inhibitor; IL-6i, the group

of patients who received IL-6 inhibitor for arthritis-irAE; TNFi, the group of patients who received TNF inhibitor for arthritis-irAE

6. Any association with HLA-B27? Any resemblance to reactive arthritis?

Answer

We investigated the presence of HLA-B27 in 19 patients (one could not be HLA-typed). Out of 19, only one patient was positive for HLA-B27.

We discuss the resemblance of arthritis-irAE to reactive arthritis below under comment #8.

In response to the reviewer's comment, we revised the supplementary table 1 as well as text (Page 7, Line 155; Page 34, Line 858-862 in the marked copy).

7. Any other arthritis mediated by IFN γ in the context of Th17? Comparisons with published literature would be useful to provide context.

Answer

We sincerely appreciate the reviewer's suggestion. Th17 cells are well known to become Th1 cells in the presence of IL-12. Indeed, the accumulation of transient (t)-Th17, and non-classical Th1 (Ex-Th17) cells in the synovium has been observed in juvenile idiopathic arthritis and RA (Maggi et al. *Frontiers in immunology*. 2019. PMID: 30930898; Kotake et al. *Journal of Clinical Medicine*. 2017. PMID: 28698517; Basdeo et al. *Journal of Immunology*. 2017. PMID: 28167631)

The reviewer 2 also raises the potential role of IFN γ in arthritis-irAE. We expanded the discussion regarding the role of IFN γ in inflammatory arthritis as well as Th17 cell plasticity in the revised manuscript (Page 18, Line 451-458; Page 21, Line 523-525 in the marked copy).

8. What is known about paraneoplastic arthritis pathogenesis? Any resemblance? Better clinical description of the arthritis cases would be very useful (polyarthritis, oligoarthritis, symmetric arthritis, etc).

Answer

There are several patterns of arthritis in arthritis-irAE as we previously described (Xerxes et al. *Current Opinion of Rheumatology*. 2019. PMID: 30870217). In this study, our patients had: undifferentiated inflammatory arthritis with oligoarthritis (n=13; mainly involving knees and/or ankles), polyarthritis (n=4; mainly involving hands and/or wrists), or monoarthritis (n=3; all unilateral knee). No patients had uveitis or enthesitis associated with arthritis.

Although our patients with arthritis-irAE developed inflammatory arthritis in the setting of malignancies, clinical features of arthritis in our cohort differs from those observed in patients with paraneoplastic arthritis, reactive arthritis, and rheumatoid arthritis. Paraneoplastic polyarthritis is a very rare form of paraneoplastic syndromes (Schett et al. *Nature Review Rheumatology*. 2014. PMID: 25136782). It can be manifested as sudden-onset polyarthritis. Autoantibodies are negative in general. One case report showed presence of one clone identical in TCR γ gene rearrangement from synovial fluid and primary tumor, which indicates potential cross reactivity between tumor antigen and self-antigen; however, detailed

mechanisms underlying paraneoplastic arthritis is unknown (Schett et al. Nature Review Rheumatology. 2014. PMID: 25136782; Schultz et al. New England Journal of Medicine. 1999. PMID: 10419392). Paraneoplastic arthritis does not respond well to immune suppressants (NSAID, steroid, and DMARDs), but the disease course is dependent on tumor burden. Patients in our cohort responded to immunosuppressants and tumor responses were favorable in general, suggesting that these were not paraneoplastic arthritis (Comments' Response Figure 11). Most of our patients (13/20) had oligoarthritis. However, they did not have extra-articular manifestations which are frequently associated with reactive arthritis such as uveitis and enthesitis. Further, only one patient was positive for HLA-B27, well-known risk allele in reactive arthritis. Although four patients in our study had rheumatoid arthritis-like polyarthritis, only one had positive rheumatoid factor. Taken together, we speculate arthritis-irAE is an inflammatory arthritis different from paraneoplastic arthritis, reactive arthritis, and rheumatoid arthritis.

In response to the reviewer's comment, we updated the supplemental table 1. Accordingly, we revised the text (Page 7; Line 151-155; Page 52-53, Table 1 in the marked copy).

9. Supplementary table 1 does not report on cutaneous side effects, either because they were not present (that would be very odd) or because they were ignored. If existing do rashes correlate with arthritis?

Answer

We thank for the reviewer's clarification comment. Five patients had CTCAE grade I (n=1) or grade II (n=4) dermatitis-irAE prior to the arthritis. None of our patients had rashes at the time of the arthritis, so we could not correlate the presence of rashes with arthritis severity.

In response to the reviewer's comment, we updated the supplemental table 1 and revised the text (Page 7; Line 155-157, Page 52-53, Table 1 in the marked copy)

10. There is much speculation and emphasis regarding mechanisms when merely based on not that strong correlative data. This aspect is more conspicuous regarding chemokines, chemokine receptors and chemoattraction.

Answer

Due to lack of pre-clinical murine models recapitulating arthritis-irAE, there are obvious limitations to investigate mechanisms. We are in the middle of generating the arthritis-irAE murine model, which will enable us to investigate the mechanisms of the arthritis-irAE more directly.

Indeed, the reviewer 2 raised the same concerns regarding chemokines/chemokine receptors, and chemoattractions. During the revision, we performed CellTrace analysis (Comments' Response Figure 4a), pseudotime analysis (Comments' Response Figure 4b), chemotaxis assay (Comments' Response Figure 5), and quantification of CXCL9/10/11/16 (Comments' Response Figure 6). We think that these data collectively support our argument that CX3CR1^{hi} CD8⁺ T cells in PB is migrated into the joint mediated by CXCL9/10/11/16 expressed by myeloid cells.

11. Would treatment with checkpoint inhibitors exacerbate collagen induced arthritis in mice? Do the model recapitulate the human findings at the molecular and cellular level? This would probably permit incisive experimentation on the matter.

Answer

We absolutely agree with the reviewer that murine models recapitulating arthritis-irAE would serve as a powerful tool in studying the mechanisms of the arthritis-irAE. Indeed, we did observed exacerbation of collagen induced arthritis (CIA), and we are generating the arthritis-irAE pre-clinical model modifying CIA. At this moment, our model recapitulate human arthritis-irAE in terms of clinical phenotypes and histology. We are in the middle of analyzing molecular/cellular characteristics in this model and compared them with human data in this manuscript.

In response to the reviewer's comment, we revised the text (Page 23, Line 573-575 in the marked copy).

Reviewers' Comments:

Reviewer #1:

Remarks to the Author:

Thorough response to my critique. Important contribution

Reviewer #2:

Remarks to the Author:

My comments have all mostly been adequately addressed.

While I appreciate the experiments done to support the hypothesis that CD8 T cell migration into the joint is mediated by CXCL9/10/11/16, I still think that the conclusion should be modified in the abstract. Instead of "is mediated" I would write "is possibly mediated". Or better to explain that several lines of evidence support the notion that the migration is mediated by these chemokines. The descriptions and updated conclusiveness for this part in the results are much appreciated. For line 489-490, I would change ", and this process is mediated by..", ", and our data strongly suggest that this process is mediated by.."

Reviewer #3:

Remarks to the Author:

With the addition of the migration assay to illustrate it is indeed CX3CR1hi CD8 T cells that are migrating in response to CXCL9/10/11/16 and the ELISA/Multiplex assays to verify CXCL9/10/11/16 levels in the SF supernatant and serum their claims are strengthened. The authors have addressed all my comments/concerns. I do not have any further comments.

Reviewer #4:

Remarks to the Author:

The revised version addresses most of my specific criticisms and those from fellow reviewers. However, there are limitations in the study, which are perhaps better acknowledged in the revised version, mainly: sample size and heterogeneity, lack of data on the tumors requiring checkpoint immunotherapy and their outcome.

RESPONSES TO REVIEWERS' COMMENTS

We thank all reviewers for their critical comments and helpful advice. We have modified the manuscript according to the recommendations. Changes are indicated by track changes in the revised manuscript. Below are the answers addressing the critiques raised by the reviewers:

REVIEWERS' COMMENTS

Reviewer #1 (Remarks to the Author):

Thorough response to my critique. Important contribution

Answer

We deeply appreciate the reviewer's valuing our findings and constructive comments throughout the review process.

Reviewer #2 (Remarks to the Author):

My comments have all mostly been adequately addressed.

While I appreciate the experiments done to support the hypothesis that CD8 T cell migration into the joint is mediated by CXCL9/10/11/16, I still think that the conclusion should be modified in the abstract. Instead of "is mediated" I would write "is possibly mediated". Or better to explain that several lines of evidence support the notion that the migration is mediated by these chemokines. The descriptions and updated conclusiveness for this part in the results are much appreciated. For line 489-490, I would change " , and this process is mediated by..", " , and our data strongly suggest that this process is mediated by.."

Answer

We sincerely thank the reviewer's insightful comments/suggestions, and certainly agree. In the abstract, the abstract, where we changed "is mediated" to "is possibly mediated" (Page 4, line 86 in the marked copy). We also revised the line 489-490 as suggested by the reviewer (Page 20, lines 495-496 in the marked copy).

Reviewer #3 (Remarks to the Author):

With the addition of the migration assay to illustrate it is indeed CX3CR1hi CD8 T cells that are migrating in response to CXCL9/10/11/16 and the ELISA/Multiplex assays to verify CXCL9/10/11/16 levels in the SF supernatant and serum their claims are strengthened. The authors have addressed all my comments/concerns. I do not have any further comments.

Answer

We genuinely appreciate the reviewer's constructive comments and suggestions during the review process.

Reviewer #4 (Remarks to the Author):

The revised version addresses most of my specific criticisms and those from fellow reviewers. However, there are limitations in the study, which are perhaps better acknowledged in the revised version, mainly: sample size and heterogeneity, lack of data on the tumors requiring checkpoint immunotherapy and their outcome.

Answer

We deeply thank for the reviewer's valuing our study and outstanding comments during the review process. We certainly agree with the reviewer and expanded the discussion of limitation of our study (Page 23, lines 563-564; Page 23, lines 577-579 in the marked copy).